# The association between different body mass index levels and midterm surgical revascularization outcomes

**Farzad Masoudkabir**[1,2]**, Negin Yavari**[1,2]**, Mana Jameie**[1,2]**, Mina Pashang**[1,2]**, Saeed Sadeghian**[1,2]**, Mojtaba Salarifar**[1]**, Arash Jalali**[1]**, Seyed Hossein Ahmadi Tafti**[1]**, Kiomars Abbasi**[1,2]**, Abbas Salehi Omran**[1]**, Shahram Momtahen**[1]**, Soheil Mansourian**[1]**, Mahmood Shirzad**[1]**, Jamshid Bagheri**[1]*****, Khosro Barkhordari**[1]**, Abbasali Karimi**[1]

**1** Tehran Heart Center, Cardiovascular Diseases Research Institute, Tehran University of Medical Sciences, Tehran, Iran, **2** Cardiac Primary Prevention Research Center, Cardiovascular Diseases Research Institute, Tehran University of Medical Sciences, Tehran, Iran

* bagheri.jamshid@gmail.com

**Data Availability Statement:** The data underlying the results presented in the study are available from Tehran Heart Center upon request. URL:

## Abstract

### Background

There are conflicting results regarding the relationship between overweight/obesity and the outcomes of coronary artery bypass graft surgery (CABG), termed "the obesity paradox". This study aimed to evaluate the effects of body mass index (BMI) on the midterm outcomes of CABG.

### Methods

This historical cohort study included all patients who underwent isolated CABG at our center between 2007 and 2016. The patients were divided into five categories based on their preoperative BMIs (kg/m$^2$): 18.5≤BMI<25, 25≤BMI<30, 30≤BMI<35, 35≤BMI<40, and BMI≥40. Patients with BMIs below 18.5 kg/m$^2$ were excluded. The endpoints of this study were all-cause mortality and major adverse cardio-cerebrovascular events (MACCEs), comprising acute coronary syndromes, cerebrovascular accidents, and all-cause mortality at five years. For the assessment of the linearity of the relationship between continuous BMI and the outcomes, plots for time varying hazard ratio of BMI with outcomes were provided.

### Results

Of 17 751 patients (BMI = 27.30 ±4.17 kg/m$^2$) who underwent isolated CABG at our center, 17 602 patients (mean age = 61.16±9.47 y, 75.4% male) were included in this study. Multivariable analysis demonstrated that patients with pre-obesity and normal weight had similar outcomes, whereas patients with preoperative BMIs exceeding 30 kg/m$^2$ kg/m$^2$ had a significantly higher risk of 5-year all-cause mortality and 5-year MACCEs than those with pre-obesity. Additionally, a positive association existed between obesity degree and all-cause mortality and MACCEs. Further, BMIs of 40 kg/m$^2$ or higher showed a trend toward higher MACCE risks (adjusted hazard ratio, 1.32; 95% confidence interval, 0.89 to 1.95), possibly

https://thc.tums.ac.ir/en/default.aspx Email:
thc@tums.ac.ir Phone: (+98) 21 88029600 to 69.

**Funding:** This study was supported by Tehran
Heart Center, Cardiovascular Diseases Research
Institute, Tehran University of Medical Sciences,
Tehran, Iran.

**Competing interests:** The authors declare that they
have no conflict of interest.

due to the small sample size. A nonlinear, albeit negligible, association was also found
between continuous BMI and the study endpoints.

## Conclusions

Our findings suggest that preoperative obesity (BMI>30 kg/m$^2$) in patients who survive early
after CABG is associated with an increased risk of 5-year all-cause mortality and 5-year
MACCEs. These findings indicate that physicians and cardiac surgeons should encourage
patients with high BMIs to reduce weight for risk modification.

## Introduction

Obesity is a rapidly growing public health concern worldwide and has emerged as the second
leading cause of death in developed countries after tobacco [1–4]. It is estimated that nearly
70% of the adult population has a body mass index (BMI, kg/m$^2$) of more than 25 kg/m$^2$,
which has increased significantly since 1980 [5, 6]. Individuals with higher BMIs are more
likely to have the major risk factors of coronary artery disease (CAD), including diabetes melli-
tus, hypertension, and hyperlipidemia. They are also more likely to develop cardiovascular dis-
eases, such as heart failure and atrial fibrillation [1, 5–8].

Parallel to the increased prevalence of obesity and CAD, higher numbers of obese patients
need to undergo coronary artery bypass graft surgery (CABG) [9–11]. There are conflicting
ideas as to whether obesity is an independent predictor of post-CABG adverse outcomes.
Numerous studies have explained the effect of "the obesity paradox", which is a reduced mor-
tality rate and perioperative morbidity in overweight or obese patients following CABG [9, 10,
12–14]. On the other hand, long-term longitudinal studies have demonstrated that obesity is
associated with more deaths in patients with cardiovascular diseases [6, 15]. Notably, a 2015
meta-analyses by Wang et al among CAD patients revealed that albeit obese patients had lower
risks of long-term mortality than normal-weight patients at a mean follow-up of 3.2 years, this
benefit of obesity vanished after 5 years of follow-up and even turned into a hazardous factor
in patients with obesity grade II/III [14]. Another recent meta-analysis with more than 865 000
pooled patients undergoing either surgical revascularization or percutaneous coronary inter-
vention (PCI) revealed that using normal weight as the reference, underweight patients suf-
fered an increased all-cause mortality risk. In contrast, the risk was lowered among overweight
(25<BMI<30), obese (30<BMI<35), and severely obese (BMI>35) patients, confirming the
concept of the obesity paradox. Interestingly, after subgroup analyses, while the paradox
remained in many subgroups, it disappeared among CABG patients, with obese and over-
weight people having nonsignificant differences from their normal-weight counterparts [6].
Another meta-analysis focusing on CABG patients showed that the odds of post-CABG mid-
to-long-term mortality were lower in overweight patients than in normal-weight patients,
although that was not the case in obese patients. In fact, the latter had a similar risk for all-
cause mortality to their normal-weight counterparts [16].

Given the conflicting results in this regard, the exact effects of obesity on midterm out-
comes after CABG still need clarification. The significant limitations of previous studies have
been their small sample size and their adjustment of regression models to a limited number of
major confounders such as age, sex, smoking, diabetes mellitus, and hypertension [17], which
may have allowed residual confounding. Therefore, it is unclear whether the so-called "obesity
paradox" is, in part, a reflection of epidemiological analyses or whether there might be

beneficial implications associated with cardiovascular outcomes among patients with obesity. Hence, this study aimed to evaluate the effects of the different BMIs on the midterm outcomes of CABG in a large cohort of patients undergoing isolated CABG.

## Material and methods

### Study population

This historical cohort study is based on our center's CABG follow-up registry, which encompasses all patients who have undergone CABG since 2007 in our center [18, 19]. All patients who underwent isolated CABG between 2007 and 2016 and survived immediately and beyond four months after surgery (the first follow-up) were enrolled in the study. Patients with BMIs below 18.5 kg/m$^2$ were excluded from the analyses as this group comprises a heterogeneous composition of frail people or, conversely, very fit people. As the study was a retrospective registry-based investigation, the institutional review board committee waived patient consent. The study protocol was approved by the Ethics Committee of Tehran Heart Center and conformed to the ethical guidelines of the Declaration of Helsinki.

### Variable definitions

The height and weight of the patients were measured at baseline, and their preoperative BMIs were calculated. BMI was defined as weight in kilograms divided by height in meters squared. The definition of the variables used in the study adhered to the guidelines of the Society of Thoracic Surgeons/the Society of Cardiovascular Anesthesiologists (STS/SCA) as follows [20].

*Hypertension* was defined by the presence of any of the following: a) history of previously diagnosed and treated hypertension; b) history of documented (on at least two occasions) systolic and/or diastolic blood pressure >140 and >90, respectively, among those without diabetes or chronic kidney disease (CKD), or history of documented systolic and/or diastolic blood pressure >130 and >80, respectively, among diabetics or those with CKD; c) current pharmacological treatment for hypertension. *Diabetes* (including type one and two, but excluding gestational diabetes or steroid-induced hyperglycemia) was defined as a history of diagnosed diabetes based on The American Diabetes Association criteria as documentation of at least one of the followings: a) hemoglobin A1c > = 6.5%; b) fasting plasma glucose > = 126 mg/dL; c) 2-h Plasma glucose > = 200 mg/dL (on glucose tolerance test); d) a random plasma glucose > = 200 mg/dL in a patient with hyperglycemia symptoms. *Hyperlipidemia* was defined as having a history of diagnosed and/or treated hyperlipidemia or having at least one of the NCEP criteria, including a) total cholesterol >200 mg/dL; b) low-density-lipoprotein cholesterol (LDL) > = 130 mg/dL; c) current treatment with anti-lipidemic medications. *Current cigarette smoking* was defined as smoking ≥100 cigarettes in total in a person who has been smoking for at least one previous month. *Opium consumption* was defined as current or former smoking or ingestion of opium. *Positive family history* of CAD was defined as the occurrence of sudden death/ PCI/ CABG/ significant coronary stenosis (>50% in at least one coronary artery) among first-degree <65 year-year-old female relatives or <55-year-old male relatives. *Cerebrovascular accidents (CVA) / transient ischemic attack (TIA)* was defined based on the patient's history or neurological consult according to patient symptoms or imaging. *COPD* was defined based on the medical history or spirometry findings (FEV1/FVC and FEV1% of predicted) pertaining to irreversible airway obstruction. *CKD* was defined as estimated glomerular filtration rate (eGFR) <60. *Prolonged* postoperative *ventilation* was defined as ventilation exceeding 24 hours. *Recent MI* was defined as MI-CABG interval <7 days.

## Follow-up protocol

According to Tehran Heart Center's follow-up protocol for post-cardiac surgery patients, the study population was invited for clinic visits at 4, 6, and 12 months after surgery and annually thereafter. Trained general practitioners visited the patients and completed a data sheet compiling data on the family history of CAD; symptoms; the major risk factors of cardiovascular diseases; the status of the control on diabetes mellitus, hypertension, hyperlipidemia, cigarette smoking, and opium abuse; laboratory and paraclinical results; and the occurrence of cardiac events (eg, acute coronary syndromes and repeat revascularization) in each visit. In the case of a patient's inability to complete a clinic visit, a telephone follow-up was completed by trained research nurses. Of 17 751 patients who underwent isolated CABG in our center, 17 602 patients (mean age: 61.16 ± 9.47 y, 75.4% male) were successfully followed (follow-up rate = 99.2%) and were included in the final analysis.

## BMI classification

The patients were categorized into 6 groups based on their baseline BMIs (kg/m$^2$) at the time of surgery: normal weight: 18.5≤BMI<25, pre-obesity: 25≤BMI<30, obesity class I: 30≤BMI<35, obesity class II: 35≤BMI≤40, and obesity class III: BMI≥40. BMI groups were categorized according to the World Health Organization (WHO) [21].

## Study endpoints

The primary endpoints were all-cause mortality and major adverse cardio-cerebrovascular events (MACCEs). MACCEs were defined as a composite of all-cause mortality, acute coronary syndromes, and/or ischemic stroke/transient ischemic attacks. No secondary endpoints were defined.

## Statistical analysis

Normally distributed continuous variables were described as the mean with the standard deviation (SD). Serum creatinine levels and intensive care unit (ICU) lengths of stay (h) were described as the median with 25th and 75th percentiles because of their skewed distributions. Categorical variables were expressed as frequencies with percentages.

Continuous variables with normal distributions were compared between the BMI groups using the one-way analysis of variance. The Kruskal–Wallis test was applied to compare serum creatinine levels and ICU lengths of stay between the BMI groups.

The unadjusted and adjusted effects of BMI on 5-year all-cause mortality and MACCEs were assessed using the Cox proportional hazards (PH) model, and the effects were reported through hazard ratios (HRs) with 95% confidence intervals (CIs). Adjustments were made on age, sex, diabetes mellitus, hypertension, hyperlipidemia, positive family history, current smoking, opium abuse, ejection fraction, chronic kidney disease, left main involvements, numbers of diseased vessels, numbers of grafts, ICU lengths of stay, off-pump vs on-pump CABG, chronic obstructive pulmonary disease, cerebrovascular accidents/transient ischemic attacks, recent myocardial infarctions, previous PCIs, and discharge medications (β-blockers, statins, aspirin/other antiplatelets, angiotensin-converting enzyme inhibitors/angiotensin II receptor blockers). Furthermore, the restricted cubic splines (RCS) with 5 knots (df:4) were applied to fit BMI on all-cause mortality and MACCEs. A time-varying HR plot was provided for the spline of continuous BMI to evaluate any possible non-linear effect between BMI on the study endpoints. Same covariates were used for adjustments as those in the Cox regression models. All the statistical analyses were conducted applying IBM SPSS Statistics for Windows, version

23.0 (Armonk, NY: IBM Corp) and Stata Statistical Software: Release 15 (College Station, TX: Stata Corp LLC).

## Results

### Population

The baseline characteristics of the study population are shown in Table 1. The distribution of preoperative BMI (mean = 27.23 ±4.24 kg/m$^2$) is depicted by histogram and is presented in supplementary materials. Those with higher BMIs were more likely to be younger than those with lower BMIs (*P*<0.001). As BMI rose, the dominance of sex was in favor of females. As was expected, there was a trend toward a higher prevalence of metabolic syndrome components, including diabetes mellitus, hypertension, and hyperlipidemia, with increasing BMIs (*Ps* for all < 0.0001). Inversely, cigarette smoking and opium abuse were significantly more frequent among those with BMIs of less than 18.5 kg/m$^2$ than those with higher BMIs (*P*<0.0001). Higher BMIs were associated with significantly higher glomerular filtration rates and significantly lower ejection fractions (*Ps* for both <0.0001).

### Follow-up

The median follow-up of the patients was 60.1 (95% CI, 59.2 to 60.9] months (maximum = 133.8 months). Moreover, 149 were completely lost to follow-up (complete follow-up rate = 99.2%). Therefore, we aimed to evaluate the 5-year outcomes of isolated CABG.

### Endpoints

MACCEs (first event) occurred in 3540 (19.9%) patients, of whom 1467 (8.3%) had acute coronary syndromes, 412 (2.3%) developed cerebrovascular accidents, and 1661 (9.4%) died from all causes. The total mortality rate was 1838 (10.4%). Fig 1 demonstrates the unadjusted hazard of all-cause mortality and MACCE among all the study cohort. Event rates among the entire study population and BMI categories are presented in S1 Table. The mortality rate of patients with BMIs of 40 kg/m$^2$ or greater (15.5%) was higher than that of the other groups by a wide margin. MACCE rates were close across the BMI groups in that they varied between 19.3% and 21.2%. A simple cumulative plot for all-cause mortality and MACCEs among all the study patients can be found in S1 Fig. The frequency of patients in the pre-obesity group was higher than that in the other groups. Based on the results of previous studies that reported lower rates of outcomes in pre-obesity BMI, we chose BMIs of between 25 kg/m$^2$ and 29.9 kg/m$^2$ as our reference category. Fig 2 illustrates the forest plot of the adjusted association between BMI categories (compared with 25≤BMI<30) and all-cause mortality and MACCEs.

Table 2 demonstrates the unadjusted and adjusted Cox regression models evaluating the effects of BMIs on 5-year all-cause mortality. Our univariate survival analysis showed significantly higher all-cause mortality rates in patients with preoperative BMIs between 18.5 kg/m$^2$ and 25 kg/m$^2$ and greater than 40 kg/m$^2$ than in those with pre-obesity (BMI = 25–29.9 kg/m$^2$). After adjustments for potential confounders, there was no significant difference in the risk of 5-year all-cause mortality between patients with pre-obesity (the reference group) and those with 18.5≤preoperative BMI<25. However, groups with BMIs of 30 kg/m$^2$ or greater had a significantly higher risk of all-cause mortality than the pre-obesity group (BMI = 25–29.9 kg/m$^2$), and a significant association was observed between the degree of obesity and all-cause mortality (Fig 3 & Table 2).

RCS analyses recapitulated the association between increasing BMI as a continuous variable (adjusted hazard ratio [aHR], 1.02; 95% CI, 1.02 to 1.05; *P*<0.001) and mortality (S2 Table). At

**Table 1. Baseline characteristics of the study population based on BMI.**

| Variables | All patients N = 17751 | 18.5≤BMI<25 n = 5547 | 25≤BMI<30 n = 8091 | 30≤BMI<35 n = 3304 | 35≤BMI<40 n = 661 | BMI≥40 n = 148 | P value |
|---|---|---|---|---|---|---|---|
| **Preoperative Characteristics** | | | | | | | |
| BMI continuous, mean (SD) | 27.30 (4.17) | 23.00 (1.52) | 27.32 (1.36) | 31.87 (1.34) | 36.80 (1.41) | 43.70 (4.58) | <0.001 |
| Age, y, mean (SD) | 61.16 (9.47) | 62.53 (9.54) | 60.95 (9.45) | 59.77 (9.20) | 59.50 (8.93) | 59.20(8.88) | <0.0001 |
| Male sex, n (%) | 13390 (75.4%) | 4590 (82.7) | 6273 (77.5) | 2168 (65.6) | 296 (44.8) | 63 (42.6) | <0.0001 |
| Diabetes, n (%) | 7148 (40.3%) | 2045 (36.9) | 3302 (40.9) | 1413 (42.8) | 312 (47.2) | 76 (51.4) | <0.0001 |
| Hypertension, n (%) | 9613 (54.2%) | 2581 (46.6) | 4346 (53.8) | 2087 (63.3) | 491 (74.5) | 108 (73.0) | <0.0001 |
| Hyperlipidemia, n (%) | 6222 (35.1%) | 1579 (28.5) | 2957 (36.5) | 1333 (40.3) | 289 (43.7) | 64 (43.2) | <0.0001 |
| Positive family History, n (%) | 6504 (37.0%) | 1874 (34.1) | 2950 (36.8) | 1343 (41.1.) | 274 (42.2) | 63 (42.9) | <0.0001 |
| GFR, median | 85.96 (67.08,107.11) | 74.30 (58.29,91.08) | 87.52 (69.78, 106.83) | 101.69 (80.24,123.9) | 111.43 (86.89,136.94) | 113.40 (86.26, 156.86) | <0.0001 |
| Left main, n (%) | 1679 (9.5%) | 587 (10.6) | 739 (9.1) | 275 (8.3) | 64 (9.7) | 14 (9.5) | 0.007 |
| Current smoking, n | 3155(17.8%) | 1177 (21.2) | 1372 (17.0) | 524 (15.9) | 73 (11.1) | 9 (6.1) | <0.0001 |
| Opium, n (%) | 2551 (14.6%) | 932 (17.1) | 1112 (13.9) | 439 (13.5) | 59 (9.0) | 9 (6.2) | <0.0001 |
| EF, mean (SD) | 47.11 (10.18) | 45.99 (10.70) | 47.3 (10.00) | 48.10 (9.72) | 48.37 (9.23) | 49.30(9.74) | <0.0001 |
| VD, n (%) | | | | | | | |
| SVD | 644 (3.7%) | 174 (3.2) | 286 (3.6) | 144 (4.4) | 32 (4.9) | 8 (5.5) | 0.008 |
| 2VD | 3782 (21.5%) | 1120 (20.4) | 1760 (22.0) | 718 (21.9) | 150 (22.8) | 34 (23.4) | |
| 3VD | 13155 (74.8%) | 4191 (76.4) | 5964 (74.5) | 2421 (73.7) | 476 (72.3) | 103 (71.0) | |
| CKD, n(%) | 2990 (16.8%) | 1523 (27.5%) | 1131 (14.0%) | 287 (8.7%) | 38 (5.7%) | 11 (7.4%) | <0.001 |
| COPD, n (%) | 628 (3.6%) | 197 (3.6) | 261 (3.2) | 132 (4.0) | 31 (4.7) | 7 (4.7) | 0.111 |
| CVA/TIA, n (%) | 1220 (6.9%) | 394 (7.1) | 568 (7.0) | 206 (6.3) | 44 (6.7) | 8 (5.4) | 0.506 |
| Previous PCI, n (%) | 812 (4.6%) | 225 (4.1) | 368 (4.5) | 179 (5.4) | 35 (5.3) | 5 (3.4) | 0.039 |
| Recent MI, n (%) | 1572 (8.9%) | 522 (9.4) | 714 (8.8) | 258 (7.8) | 62 (9.4) | 16 (10.8) | 0.110 |
| **Intraoperative/Postoperative Characteristics** | | | | | | | |
| Perfusion time, mean (SD) | 68(55,85) | 67(55,84) | 67(54,83) | 69(55,85) | 70(55,85) | 70(60,90) | 0.079 |
| Cross-clamp time, mean (SD) | 39(30,50) | 39(30,49) | 39 (30,50) | 40 (30,50 | 40 (32,50) | 40 (32,52) | 0.012 |
| Off-pump, n (%) | 1526 (8.6%) | 468 (8.4) | 671 (8.3) | 293 (8.9) | 71 (10.07) | 18 (12.2) | 0.074 |
| Graft number, median | 3 (3, 4) | 3 (3,4) | 3 (3,4) | 3 (3,4) | 3 (3,4) | 3 (3,4) | 0.002 |
| Arterial grafts, n (%) | | | | | | | 0.001 |
| None | 186(1.0%) | 87 (1.6%) | 67 (0.8%) | 28 (0.8%) | 3 (0.5%) | 1(0.7%) | |
| One | 17215 (97.0%) | 5354(96.5%) | 7854 (97.1%) | 3212(97.2) | 650(98.3%) | 145(98.0%) | |
| Two/three | 350 (2.0%) | 106 (1.9%) | 170 (2.1%) | 64 (1.9%) | 8(1.2%) | 2 (1.4%) | |
| Venous grafts, median | 2 (2,3) | 2 (2,3) | 2 (2,3) | 2 (2,3) | 2 (2,3) | 2 (2,3) | 0.001 |
| ICU hours, median | 29 (23,65) | 40 (23, 67.5) | 28.5 (23, 53.5) | 27.50 (22.5, 66) | 27.50 (23, 66) | 40.25 (23.15, 71.75) | <0.0001 |
| IMA, n(%) | | | | | | | 0.001 |
| Left IMA | 17431 (98.2%) | 5416(97.6%) | 7955 (98.3%) | 3256(98.5 | 658 (99.5%) | 146 (98.6%) | |
| Right IMA | 17 (0.1%) | 4 (0.1%) | 7 (0.1%) | 6 (0.2%) | 0 | 0 | |
| Both | 107 (0.6%) | 35 (0.6%) | 54 (0.7%) | 17 (0.5%) | 0 | 1 (0.7%) | |
| None | 196 (1.1%) | 92 (1.7%) | 75 (0.9%) | 25 (0.8%) | 3 (0.5%) | 1 (0.7%) | |
| Urgent/emergent surgery, n (%) | 296 (1.7%) | 94 (1.7%) | 137 (1.7%) | 54 (1.6%) | 10 (1.5%) | 1 (0.7%) | 0.899 |
| Perioperative IABP, n(%) | 281 (1.6%) | 96 (1.7%) | 125 (1.6%) | 47 (1.4%) | 13 (2.0%) | 0 | 0.353 |
| **Postoperative Complications, n (%)** | | | | | | | |
| ICU blood transfusion | 5022 (28.4%) | 1776(32.1%) | 2181 (27.1%) | 827 (25.1%) | 193 (29.4%) | 45 (30.6%) | <0.001 |
| CVA/TIA | 147 (0.8%) | 49 (0.9%) | 60 (0.7%) | 31 (0.9%) | 7 (1.1%) | 0 | 0.542 |

*(Continued)*

**Table 1.** (Continued)

| Variables | All patients N = 17751 | 18.5≤BMI<25 n = 5547 | 25≤BMI<30 n = 8091 | 30≤BMI<35 n = 3304 | 35≤BMI<40 n = 661 | BMI≥40 n = 148 | P value |
|---|---|---|---|---|---|---|---|
| Prolonged ventilation | 375 (2.1%) | 150 (2.7%) | 142 (1.8%) | 63 (1.9%) | 18 (2.7%) | 2 (1.4%) | 0.002 |
| Reoperation for bleeding/ tamponade | 431 (2.4%) | 172 (3.1%) | 187 (2.3%) | 59 (1.8%) | 10 (1.5%) | 3 (2.0%) | 0.001 |
| **Discharge medications, n (%)** | | | | | | | |
| ACEI/ARB | 7790 (44.0%) | 2212(40.0%) | 3537 (43.8%) | 1625(49.4%) | 344 (52.2%) | 72 (49.0%) | <0.001 |
| B-blockers | 16531 (93.2%) | 5138(92.8%) | 7575 (93.7%) | 3075(93.1%) | 609 (92.1%) | 134(91.2%) | 0.168 |
| Statins | 16738 (94.4%) | 5228(94.4%) | 7651 (94.6%) | 3103(94.0%) | 618 (93.5%) | 138(93.9%) | 0.599 |
| ASA/ antiplatelets | 17154 (96.7%) | 5346(96.6%) | 7851 (97.1%) | 3187(96.5%) | 632 (95.6%) | 138(93.9%) | 0.030 |

BMI, Body mass index; SD, Standard deviation; GFR, Glomerular filtration rate; EF, Ejection fraction; VD, Vessel disease; SVD, Single-vessel disease; MI, Myocardial infarction; COPD, Chronic obstructive pulmonary disease; CVA, Cerebrovascular accidents; TIA, Transient ischemic attack; PCI, Percutaneous coronary intervention; ICU, Intensive care unit; IMA, Internal mammary artery; IAPB, Intra-aortic balloon pump; ACEI, Angiotensin-converting enzyme inhibitor; ARB, Angiotensin II receptor blocker; ASA, Aspirin.

* Continuous variables are presented as the mean (SD) or the median (25th and 75th percentiles).

* Categorical variables are described as frequencies (percentages); n (%).

univariable level, the risk of 5-year MACCEs was significantly higher in patients with BMIs (kg/m$^2$) of less than 18.5, between 18.5 and 24.9, and between 30 and 34.9 than in the pre-obesity group. Nonetheless, our univariate analysis detected similar risks of MACCEs between all the groups with BMIs of greater than 35 kg/m$^2$ and the pre-obesity group (Table 3). After adjustments for the aforementioned potential confounders, a significant association was

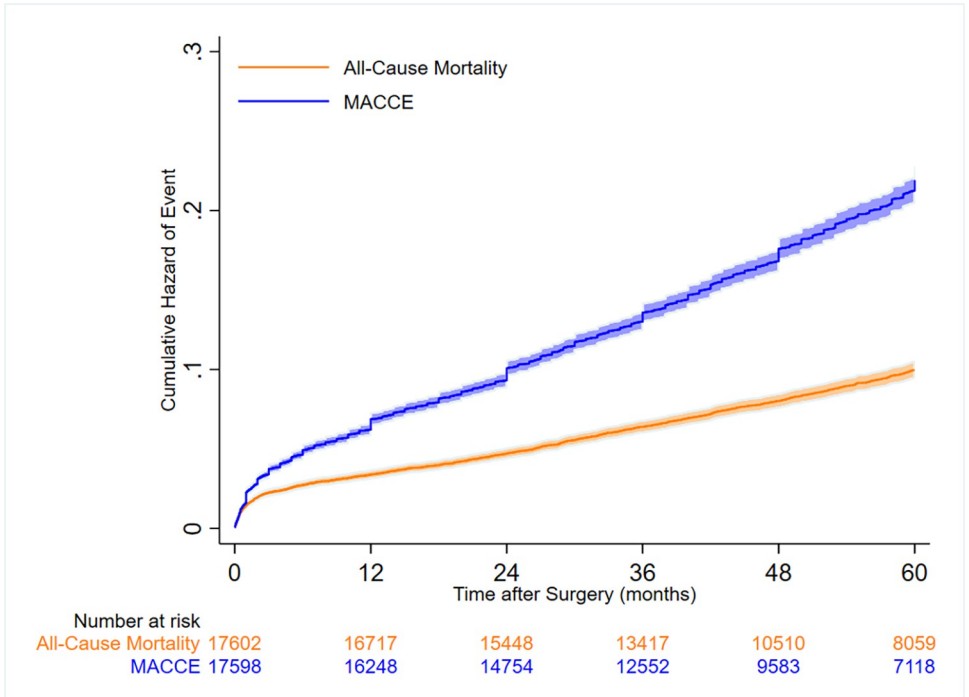

**Fig 1. Unadjusted cumulative hazard of all-cause mortality and major adverse cardio-cerebrovascular events (MACCEs) after coronary artery bypass graft surgery (CABG) among all the study cohort.**

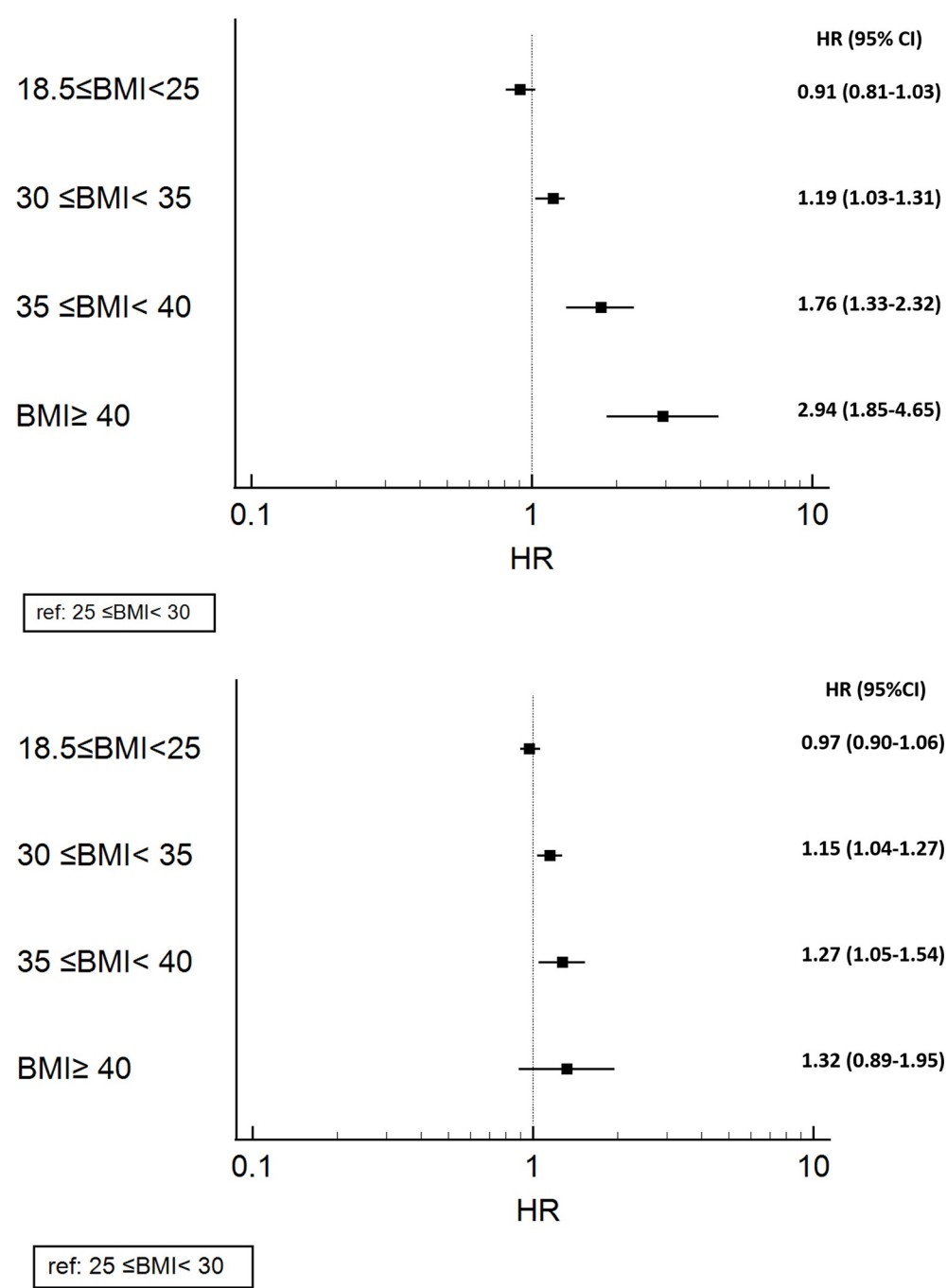

**Fig 2. Forest plot of the adjusted effects of different levels of body mass index (BMI) on all-cause mortality and major adverse cardio-cerebrovascular events (MACCEs) after coronary artery bypass graft surgery (CABG).**

observed between the degree of obesity (from BMI>30 kg/m$^2$) and the risk of 5-year MACCEs (Fig 4 & Table 3).

The group with BMIs of 40 kg/m$^2$ or higher, with the highest MACCE aHR, showed a trend toward increasing risks (aHR, 1.32; 95% CI, 0.89 to 1.95; $P$ = 0.161), probably caused by the small sample size of this group compared with the others. This association was supported by findings from the RCS analyses (aHR, 1.02; 95% CI, 1.01 to 1.03; $P$<0.001) (S3 Table).

**Table 2. Effects of the different levels of BMI on 5-year mortality after isolated coronary artery bypass graft surgery.**

| Variables | HR | 95% CI | P value | Global P value |
|---|---|---|---|---|
| **Unadjusted** | | | | |
| 25 ≤BMI< 30* | | | | <0.0001 |
| 18.5 ≤BMI< 25 | 1.22 | 1.01–1.35 | <0.0001 | |
| 30 ≤BMI< 35 | 1.01 | 0.89–1.16 | 0.836 | |
| 35 ≤BMI< 40 | 1.08 | 0.84–1.34 | 0.533 | |
| BMI≥ 40 | 1.87 | 1.24–2.84 | 0.003 | |
| **Adjusted**** | | | | |
| 25 ≤BMI< 30* | | | | <0.0001 |
| 18.5 ≤BMI< 25 | 0.91 | 0.81–1.03 | 0.126 | |
| 30 ≤BMI< 35 | 1.19 | 1.03–1.31 | 0.016 | |
| 35 ≤BMI< 40 | 1.76 | 1.33–2.32 | <0.0001 | |
| BMI≥ 40 | 2.94 | 1.85–4.65 | <0.0001 | |
| Age, y | 1.05 | 1.05–1.06 | <0.0001 | |
| Male sex | 2.56 | 2.19–2.98 | <0.0001 | |
| Diabetes | 1.46 | 1.31–1.62 | <0.0001 | |
| Hypertension | 1.36 | 1.22–1.51 | <0.0001 | |
| Hyperlipidemia | 0.91 | 0.81–1.02 | 0.119 | |
| Family history | 0.99 | 0.89–1.1 | 0.829 | |
| CKD | 1.72 | 1.52–1.94 | <0.0001 | |
| Left main | 0.97 | 0.82–1.15 | 0.733 | |
| Current smoking | 1.21 | 1.05–1.40 | 0.01 | |
| Opium | 1.22 | 1.05–1.41 | 0.01 | |
| EF | 0.97 | 0.96–0.97 | <0.0001 | |
| SVD | | | | |
| 2VD | 0.82 | 0.59–1.15 | 0.247 | |
| 3VD | 1.13 | 0.82–1.56 | 0.463 | |
| Graft number | 0.90 | 0.84–0.96 | 0.001 | |
| ICU hours ‡ | 1.001 | 1.001–1.001 | <0.0001 | |
| Off pump | 1.07 | 0.87–1.32 | 0.535 | |
| Recent MI | 0.99 | 0.83–1.17 | 0.887 | |
| COPD | 1.28 | 1.02–1.61 | 0.031 | |
| CVA/TIA | 1.53 | 1.30–1.80 | <0.0001 | |
| Previous PCI | 0.96 | 0.74–1.25 | 0.753 | |
| Discharge medications | | | | |
| ACEI/ARB | 0.96 | 0.87–1.07 | 0.486 | |
| ASA/antiplatelets | 0.39 | 0.32–0.47 | <0.001 | |
| Statins | 0.46 | 0.39–0.54 | <0.001 | |
| B-blockers | 0.56 | 0.48–0.65 | <0.001 | |

HR; Hazard ratio; CI, Confidence interval; BMI, Body mass index; CKD, Chronic kidney disease; EF, Ejection fraction; SVD, Single-vessel disease; VD, Vessel disease; ICU, Intensive care unit; MI, Myocardial infarction; COPD, Chronic obstructive pulmonary disease; CVA Cerebrovascular accidents; TIA, Transient ischemic attack; PCI, Percutaneous coronary intervention; ACEI, Angiotensin-converting enzyme inhibitor; ARB, Angiotensin II receptor blocker; ASA, Aspirin.

‡ Per 10-hour increase.

* Reference category.

**Adjusted for age, sex, diabetes mellitus, hypertension, hyperlipidemia, family history, current smoking, opium abuse, CKD, ejection fractions, left main involvements, numbers of diseased vessels, numbers of grafts, intensive care unit stay, off-pump surgery, myocardial infarction of less than 7 days, chronic obstructive pulmonary disease, cerebrovascular disease, previous percutaneous coronary interventions, discharge medications (β-blockers, statins, aspirin or other anti-platelets, angiotensin-converting enzyme inhibitors [ACEIs]/angiotensin II receptor blockers [ARBs]).

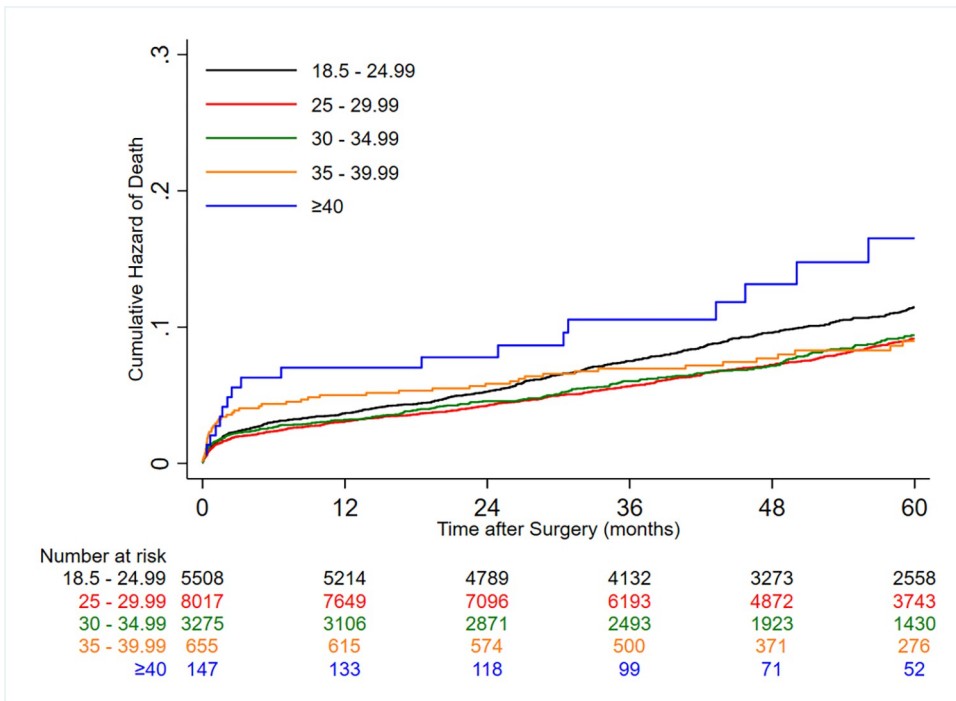

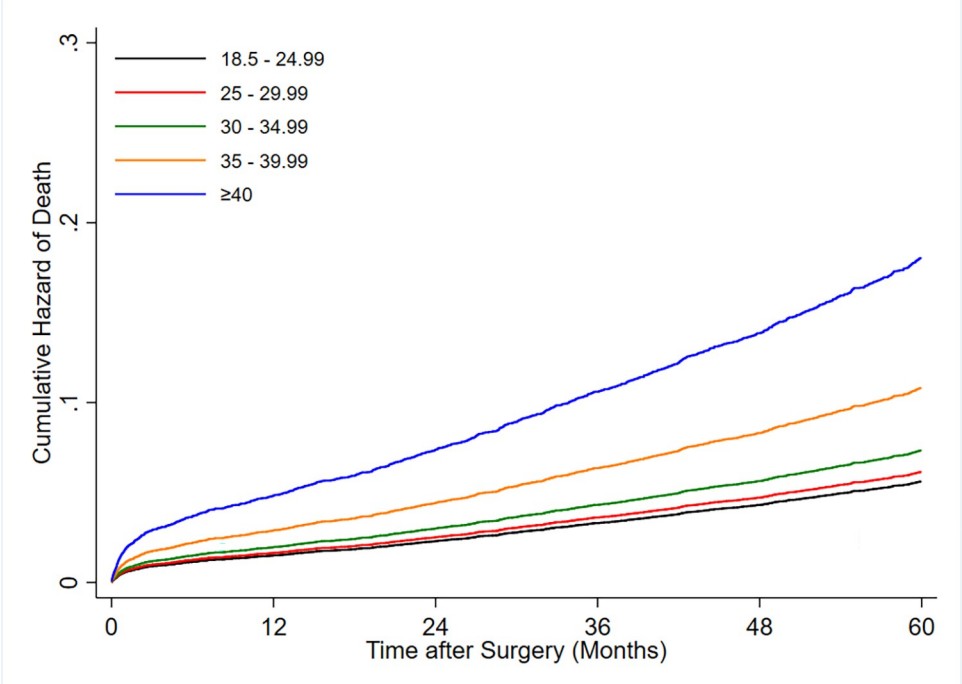

**Fig 3.** (a) Unadjusted and (b) adjusted cumulative hazard of all-cause mortality after coronary artery bypass graft surgery (CABG) according to different levels of body mass index (BMI).

Meanwhile, once again, BMIs between 18.5 and 24.9 were not associated with an increased risk of MACCEs by comparison with the reference group.

Plots for the time-varying hazard ratio of BMI with study outcomes are depicted in Fig 5. There were some minuscule variations (≈<2%) in the HR plots; therefore, the association

**Table 3. Effects of different levels of BMI on 5-year MACCEs after isolated coronary artery bypass graft surgery.**

| Variables | HR | 95% CI | P value | Global P value |
|---|---|---|---|---|
| **Unadjusted** | | | | |
| 25 ≤BMI< 30* | | | | 0.011 |
| 18.5 ≤BMI< 25 | 1.10 | 1.02–1.20 | 0.010 | |
| 30 ≤BMI< 35 | 1.10 | 1.01–1.21 | 0.029 | |
| 35 ≤BMI< 40 | 1.12 | 0.94–1.35 | 0.206 | |
| BMI≥ 40 | 1.24 | 0.87–1.77 | 0.228 | |
| **Adjusted**** | | | | |
| 25 ≤BMI< 30* | | | | <0.0001 |
| 18.5 ≤BMI< 25 | 0.97 | 0.90–1.06 | 0.544 | |
| 30 ≤BMI< 35 | 1.15 | 1.04–1.27 | 0.006 | |
| 35 ≤BMI< 40 | 1.27 | 1.05–1.54 | 0.014 | |
| BMI≥ 40 | 1.32 | 0.89–1.95 | 0.161 | |
| Age | 1.02 | 1.01–1.02 | <0.0001 | |
| Male | 1.27 | 1.16–1.39 | <0.0001 | |
| Diabetes | 1.29 | 1.20–1.39 | <0.0001 | |
| Hypertension | 1.29 | 1.19–1.39 | <0.0001 | |
| Hyperlipidemia | 0.92 | 0.85–1.00 | 0.038 | |
| Family history | 1.02 | 0.94–1.09 | 0.675 | |
| CKD, n(%) | 1.44 | 1.31–1.57 | <0.0001 | |
| Left main | 0.96 | 0.85–1.09 | 0.555 | |
| Current smoking | 1.10 | 0.99–1.22 | 0.083 | |
| Opium | 1.11 | 0.99–1.23 | 0.066 | |
| EF | 0.98 | 0.98–0.98 | <0.0001 | |
| SVD | | | | |
| 2VD | 0.87 | 0.70–1.08 | 0.212 | |
| 3VD | 1.01 | 0.82–1.24 | 0.943 | |
| Graft number | 0.93 | 0.89–0.97 | 0.001 | |
| ICU hours ‡ | 1.001 | 1.001–1.001 | <0.0001 | |
| Off-pump | 1.00 | 0.86–1.16 | 0.985 | |
| Recent MI | 1.01 | 0.90–1.15 | 0.831 | |
| COPD | 1.15 | 0.97–1.37 | 0.11 | |
| CVA/TIA | 1.40 | 1.23–1.58 | <0.0001 | |
| Previous PCI | 1.26 | 1.07–1.48 | 0.006 | |
| Discharge Medications | | | | |
| ACEI/ARB | 1.01 | 0.91–1.09 | 0.778 | |
| ASA/antiplatelets | 0.5 | 0.43–0.58 | <0.001 | |
| Statins | 0.61 | 0.54–0.7 | <0.001 | |
| B-blockers | 0.70 | 0.62–0.78 | <0.001 | |

MACCE, Major cardio-cerebrovascular events; HR, Hazard ratio; CI, Confidence interval; BMI, Body mass index; CKD, Chronic kidney disease; EF, Ejection fraction; SVD, Single-vessel disease; VD, Vessel disease; ICU, Intensive care unit; MI, Myocardial infarction; COPD, Chronic obstructive pulmonary disease; CVA, Cerebrovascular accidents; TIA, Transient ischemic attack; PCI, Percutaneous coronary intervention; ACIE, Angiotensin-converting enzyme inhibitor; ARB, Angiotensin II receptor blocker; ASA, Aspirin.

‡ Per 10-hour increase.

* Reference category.

** Adjusted for age, sex, diabetes mellitus, hypertension, hyperlipidemia, family history, current smoking, opium abuse, CKD, ejection fractions, left main involvements, numbers of diseased vessels, numbers of grafts, intensive care unit stay, off-pump surgery, myocardial infarction of less than 7 days, chronic obstructive pulmonary disease, cerebrovascular disease, previous percutaneous coronary interventions, discharge medications (β-blockers, statins, aspirin or other anti-platelets, angiotensin-converting enzyme inhibitors [ACEIs]/angiotensin II receptor blockers [ARBs]).

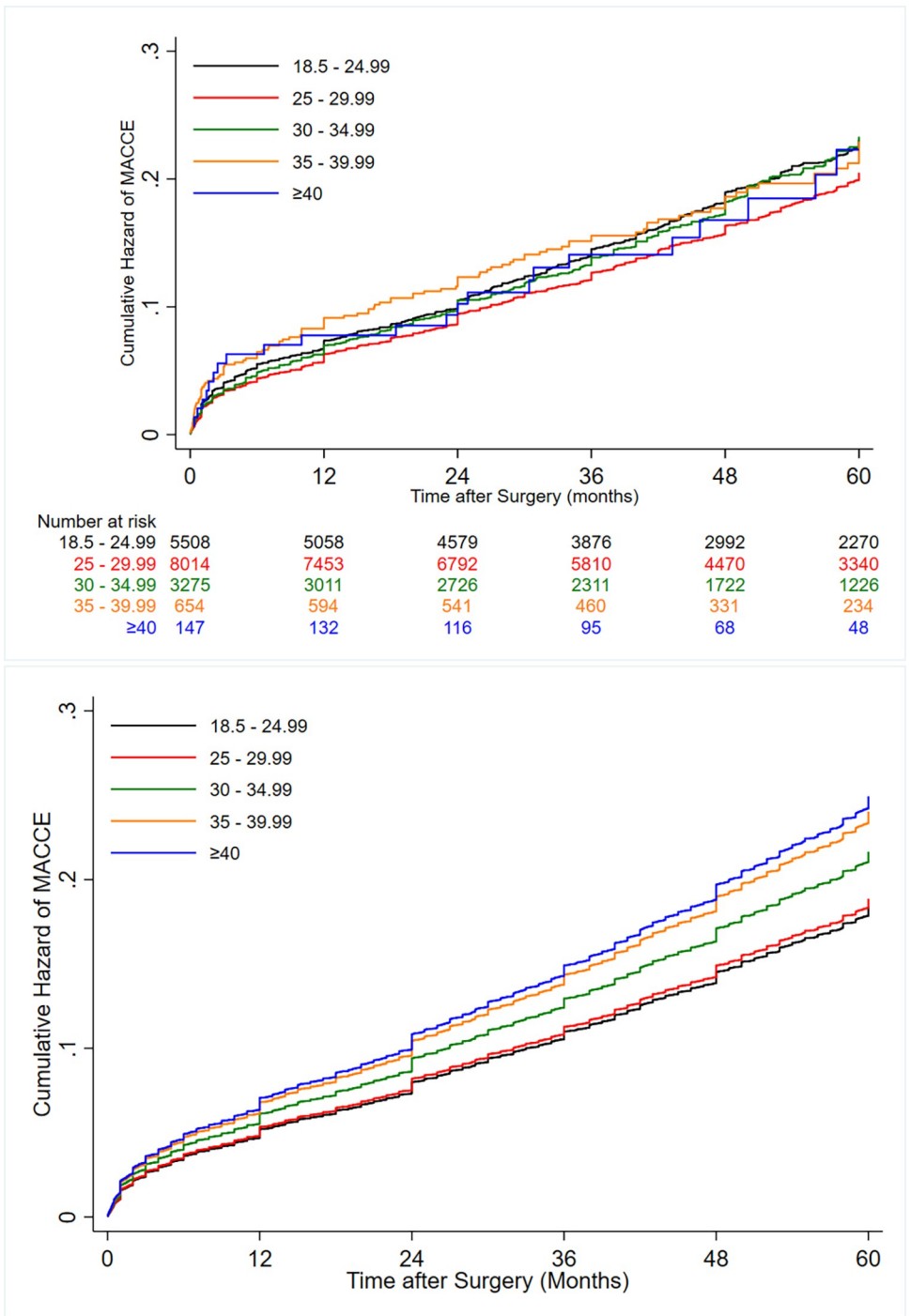

**Fig 4.** (a) Unadjusted and (b) adjusted cumulative hazard of major adverse cardio-cerebrovascular events (MACCEs) after coronary artery bypass graft surgery (CABG) according to different levels of body mass index (BMI).

between BMI and mortality and MACCEs seemed to be nonlinear, albeit negligibly (considering the HR scales). These findings confirmed that the evaluation of BMI as a categorical variable did not distort the findings from conventional Cox models; specially the points at which the curve's slope changed were almost compatible with this study's BMI cutoff points.

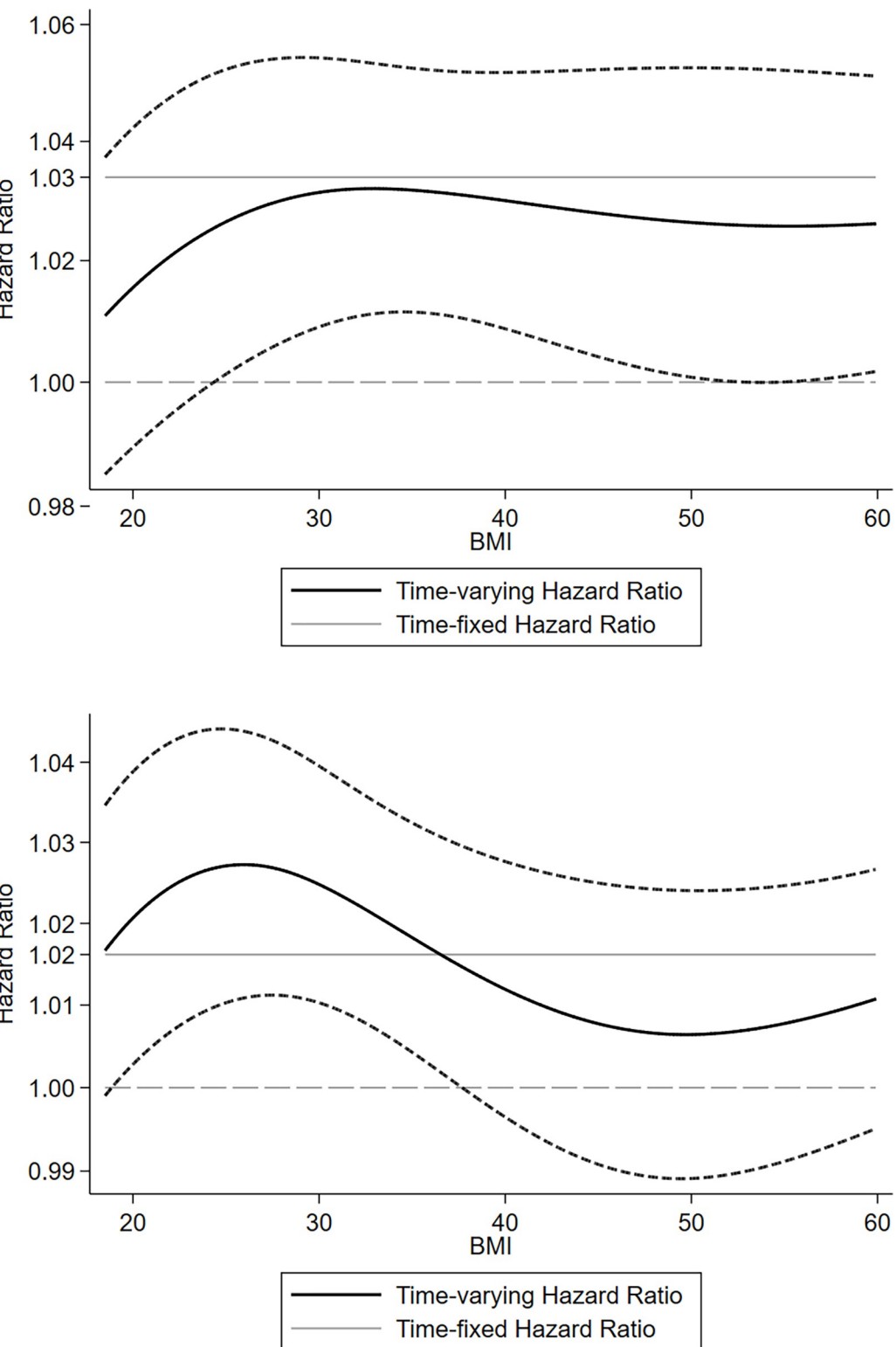

**Fig 5.** Time varying hazard ratio of BMI and (a) all-cause mortality and (b) major adverse cardio-cerebrovascular events (MACCEs).

## Discussion

### Major findings

The major finding of our study was a worse midterm prognosis in patients undergoing isolated CABG with BMIs of higher than 30 kg/m$^2$. Additionally, we found a significant positive association between the degree of obesity according to BMI and a higher midterm risk of all-cause mortality MACCEs. Our results also showed a nonlinear, albeit negligible, association between continuous BMI and the study endpoints.

### Impact of BMI on outcomes: Current literature

Several studies in the past have demonstrated paradoxically better clinical outcomes for pre-obesity and obesity patients and CAD compared with normal-weight patients, while other studies have questioned this association, triggering a hot debate. After the first description of the paradoxical survival advantage of patients with pre-obesity versus normal weight following PCI by Gruberg et al. [22], other studies reported similar results in atrial hypertension, heart failure, diabetes mellitus, and post-revascularization procedures such as PCI and CABG [23–25]. The relationship between BMI and the surgical revascularization outcome is complex, with multiple studies reporting better [7, 22, 26–29], similar [30–32], or worse [12, 33–36] outcomes in patients with obesity [36, 37].

In contrast to our findings, several studies on patients undergoing CABG have shown more favorable short-term and midterm survival rates for patients with higher levels of BMI, implying a possible protective effect for obesity, termed "the obesity paradox" [7, 22, 26, 27]. The APPROACH Registry followed up 7617 patients who underwent CABG between the years 2001 and 2006 for a median of 46 months and reported that in the CABG group, BMIs of between 30.0 kg/m$^2$ and 34.9 kg/m$^2$ had the lowest risk of mortality (aHR, 0.75; 95% CI, 0.61 to 0.94) [29]. Notably, the APPROACH Registry recruited patients between the years 2001 and 2006, which is about a decade earlier than our cohort of patients recruited between the years 2007 and 2016. Indubitably, surgical techniques, the medical management of patients, and the rate of prescription and doses (particularly concerning statins) have undergone drastic changes over time, which could justify the difference in findings between our more recent study and the APPROACH Registry, at least in part. More importantly, the APPROACH Registry was created to register patients with established CAD with all types of treatment modalities, including medical treatment, PCI, and CABG. Further, the authors of that registry failed to report the variables for which they adjusted their models. Accordingly, we suspect that they might not have been able to adjust their models for a variety of the major predictors of outcomes in patients undergoing CABG, and their findings are likely to suffer from residual confounding.

In line with our findings, a long-term follow-up of 1526 patients who underwent CABG in the BARI Trial showed that 5-year mortality was nearly 5-fold higher in patients with higher BMIs (adjusted RR, 4.86; $P$ = 0.01) than in those with normal BMIs (adjusted RR, 1.0) [36]. Van Straten et al investigated the effects of BMI on 10 268 patients after CABG and found that overweight failed to confer survival advantages and that morbid obesity was an independent risk factor for late mortality [12]. Several explanations may account for the discrepancies observed in our study and those in favor of the obesity paradox. A potential explanation may be the confounding effects of major predictors such as smoking. Smoking patients are known to have lower BMIs [38, 39], while they carry a high risk of short- and long-term adverse cardiac events after CABG [40, 41]. In this study, we performed comprehensive adjustments for many potential confounders. Obesity was previously considered a perioperative risk factor in

CABG [42]. It is, therefore, possible that in previous studies, high-risk obese patients were excluded from revascularization, creating a selection bias. In other words, patients with higher BMIs had better survival due to better risk profiles, in particular, younger age.

## Clinical implications

The clear message of our study is that in patients who survive early after cardiac surgery, pre-obesity confers no advantages over normal BMI, patients with obesity (BMI>30 kg/m$^2$) are at an increased risk of 5-year all-cause mortality and 5-year MACCEs, and there is a significant positive association between the degree of obesity and 5-year risks of all-cause mortality and MACCEs. These findings indicate that the obesity paradox is not applicable to patients undergoing isolated CABG, and physicians and cardiac surgeons should implement risk modification to encourage patients with BMIs of higher than 30 kg/m$^2$ to reduce weight and decrease the risk of midterm adverse events. It is also important for future studies to further evaluate the impact of weight reduction on the long-term survival of obese patients undergoing CABG. It is also imperative to notice that obesity is associated with a high rate of cardiovascular and non-cardiovascular morbidity and mortality. It is unlikely that coronary revascularization completely attenuates this increased risk, hence the importance of stricter surveillance in this group of patients in order to alleviate poorer outcomes.

## Study strengths and limitations

To our knowledge, our investigation is the largest study of its kind to feature a long-term follow-up on patients undergoing isolated CABG. The large sample size and the duration of follow-up enabled us to detect differences between BMI categories. Moreover, in contrast to registries like APPROACH, our investigation was specifically designed for patients undergoing CABG, and not only did it include data on the major risk factors of CAD but also it collected intraoperative and perioperative data, allowing comprehensive adjustments for potential confounders. Additionally, our cohort of patients is the most contemporary cohort to undergo CABG and is more compatible with the current real-world management and outcomes of CABG. Still, we failed to evaluate our patients using waist circumference and body composition analysis and, similar to other relevant studies, based our analyses on preoperative BMI evaluations. We also failed to measure BMI during the follow-up, which is liable to significant changes over time. In addition, we had considerable missing data on peripheral arterial diseases, which constitute a significant risk factor, precluding us from integrating this risk factor into our analytic models.

## Supporting information

**S1 Table. Event rates in the study cohort and by BMI category.** BMI, body mass index; MACCE, major adverse cardio-cerebrovascular events; ACS, acute coronary syndrome; CVA, cerebrovascular events. Data are presented as number and frequency.
(DOCX)

**S2 Table. The RCS model on All-cause mortality.** HR, Hazard ratio; CI, Confidence interval; BMI, Body mass index; CKD, Chronic kidney disease; EF, Ejection fraction; SVD, Single vessel disease; VD, Vessel disease; ICU, Intensive care unit; MI, Myocardial infarction; COPD, Chronic obstructive pulmonary disease; CVA, Cerebrovascular accidents; TIA, Transient ischemic attack; PCI, Percutaneous coronary intervention; ACIE, Angiotensin converting enzyme inhibitor; ARB, Angiotensin II receptor blocker; ASA, Aspirin.
(DOCX)

**S3 Table. The RCS model on MACCE.** MACCE, Major cardio-cerebrovascular events; HR, Hazard ratio; CI,Confidence interval; BMI, Body mass index; CKD, Chronic kidney disease; EF, Ejection fraction; SVD,Single vessel disease; VD, Vessel disease; ICU, Intensive care unit; MI, Myocardial infarction; COPD, Chronic obstructive pulmonary disease; CVA, Cerebrovascular accidents; TIA, Tranisent ischemic attack; PCI, Percutaneous coronary intervention; ACIE, Angiotensin converting enzyme inhibitor,; ARB, Angiotensin II receptor blocker; ASA, Aspirin.
(DOCX)

**S1 Fig. BMI distribution among the study cohort.**
(TIF)

## Acknowledgments

We would like to appreciate all patients and the hospital staff for their valuable continuation to this study.

## Author Contributions

**Conceptualization:** Farzad Masoudkabir, Negin Yavari, Saeed Sadeghian, Mojtaba Salarifar, Seyed Hossein Ahmadi Tafti, Kiomars Abbasi, Shahram Momtahen, Soheil Mansourian, Mahmood Shirzad, Jamshid Bagheri, Khosro Barkhordari, Abbasali Karimi.

**Data curation:** Negin Yavari, Mana Jameie, Mina Pashang, Arash Jalali, Abbas Salehi Omran, Soheil Mansourian.

**Formal analysis:** Mana Jameie, Mina Pashang, Arash Jalali.

**Investigation:** Saeed Sadeghian, Mojtaba Salarifar, Kiomars Abbasi, Abbas Salehi Omran, Shahram Momtahen, Soheil Mansourian, Abbasali Karimi.

**Methodology:** Farzad Masoudkabir, Negin Yavari, Arash Jalali, Seyed Hossein Ahmadi Tafti, Mahmood Shirzad, Jamshid Bagheri, Khosro Barkhordari.

**Project administration:** Farzad Masoudkabir, Jamshid Bagheri.

**Supervision:** Farzad Masoudkabir, Saeed Sadeghian, Jamshid Bagheri.

**Validation:** Saeed Sadeghian, Mojtaba Salarifar, Arash Jalali, Seyed Hossein Ahmadi Tafti, Kiomars Abbasi, Abbas Salehi Omran, Jamshid Bagheri.

**Writing – original draft:** Negin Yavari, Mana Jameie, Mina Pashang, Shahram Momtahen.

**Writing – review & editing:** Farzad Masoudkabir, Saeed Sadeghian, Mojtaba Salarifar, Arash Jalali, Seyed Hossein Ahmadi Tafti, Kiomars Abbasi, Abbas Salehi Omran, Shahram Momtahen, Soheil Mansourian, Mahmood Shirzad, Jamshid Bagheri, Khosro Barkhordari, Abbasali Karimi.

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
