## [Decision Letter · Decision Letter 0]

4 May 2022

PONE-D-22-09560Effects of Different Body Mass Index Levels on Long-term Surgical Revascularization OutcomesPLOS ONE

Dear Dr. Bagheri,

Thank you for submitting your manuscript to PLOS ONE. After careful consideration, we feel that it has merit but does not fully meet PLOS ONE’s publication criteria as it currently stands. Therefore, we invite you to submit a revised version of the manuscript that addresses the points raised during the review process. Your paper is on an important topic. Although this has been addressed many times, PLOS One does not grade papers on novelty, only on scientific integrity and conduct of research. Therefore, we are providing you with the opportunity to revise your paper and we hope that you can address all the comments of the editor/reviewers. 

We look forward to receiving your revised manuscript.

Kind regards,

Salil Deo

Academic Editor

PLOS ONE

Journal Requirements:

"The authors received no financial support for the research, authorship, and/or publication of this article. "

Additional Editor Comments:

This is a retrospective study that studies the association between preoperative BMI and 5-year outcomes after CABG.

I have reviewed the study too. Along with the other reviewers, please see and consider my comments below –

1.Please expand on the introduction and state what you aim to add in your study. Some actual results from prior studies would be good to report.

2. There are many issues in the methods section. Authors need to provide very clear definitions.

Please explain definitions for other variables in the study. DM - only type 2 or all, what were the baseline medications prior to surgery, please state and rephrase your observed events as primary and secondary endpoints , why opium use, is it very prevalent in your area ?

3. For table 1, please provide an overview of all patients and then according to category of BMI. BMI groups are according to the WHO. Please state that in the methods section. It would also be good to report BMI and a continuous variable and maybe provide a histogram of distribution.

4. What was your median and maximum follow-up period ? Both need to be reported. Please report the event rates per category first and then further analyses. Results may not be very robust as there are only 141 patients in the < 18.5 group. I would recommend combining < 18.5 & 18.5 – 25 groups. For your study, I would in fact remove < 18.5. You have very few patients in this group and they may be frail people, or very fit people which are both different category of patients.

5. Please state clearly all the variables used for adjustment in the model. I would recommend that authors use a spline term to fit BMI and also present results of that regression model.

6. Please provide adjusted analyses results in a separate table, or a forest plot would be better.

7. Figures – please provide a simple cumulative plot for all-cause mortality and MACCE for the whole group with confidence intervals. Then provide figures for each BMI category. I would prepare separate plots for each BMI category and provide confidence intervals and # patients at risk for each time point listed on the x axis. It would be good to also see a HR plot for the spline of BMI as a continuous variable to see if the increase in HR is nonlinear for increasing BMI.

8. As reviewer states, the title provides a causal link, but this is a paper looking at association not causation. – change the title please to remove this causal language.

9. In the abstract and paper, you need to clearly state that this is ‘preoperative BMI’ and again provide BMI as a continuous variable before splitting it into groups.

10. There is no mention regarding medications that patients are on. These should be used to adjust for in the model. Some variables like ICU stay, opium use, are not very meaningful for 5-year outcomes and can be removed from the model. Rather than graft # and number of diseased vessels, complete vs incomplete revascularization would be better. MI under 7 hours can be changed to recent MI. eGFR can be changed to CKD with CKD – eGFR < 60. That would be more clinically meaningful.

11. Rather than only considering BMI, can authors also combine BMI, DM and hyperlipidemia to identify those with metabolic syndrome and also present results for patients with and without metabolic syndrome.

12. presence of PAD is very important as a risk factor and should be reported in table 1 and included in the Cox model.

13. I would not consider 5 years to be long term for CABG outcomes; long term for CABG would be 10 years and beyond. Please change long term to mid-term.

14. Please restructure the discussion as follows –

P1 = what we have observed

P2 = current literature and how what we have found is the same or different / why if different ?

P3 = clinical implications of our findings

P4 = Strength and limitations.

15. Data on follow-up was collected by visits. Do you have BMI at follow up and can you model change in BMI and outcomes? Most papers only look at preoperative BMI and change in BMI would be very interesting to see.

Reviewers' comments:

Reviewer's Responses to Questions

**Comments to the Author**

1. Is the manuscript technically sound, and do the data support the conclusions?

Reviewer #1: Yes

Reviewer #2: Yes

2. Has the statistical analysis been performed appropriately and rigorously? 

Reviewer #1: Yes

Reviewer #2: Yes

3. Have the authors made all data underlying the findings in their manuscript fully available?

Reviewer #1: Yes

Reviewer #2: Yes

4. Is the manuscript presented in an intelligible fashion and written in standard English?

Reviewer #1: Yes

Reviewer #2: Yes

5. Review Comments to the Author

Reviewer #1: This study does represent an interesting topic in obesity paradox. This study does have clinical priority however there exist many ways in which an unobserved covariate or several various factor lead to confounding to explain results. As discussed in study, BMI was not serially monitored in the patients in follow-up given the categories the patients were stratified to which may inappropriately bias them into a cohort. I believe the title is misleading as their is no effect of BMI on revascularization outcomes but found associations in particular to obese individuals.

Reviewer #2: This is a retrospective observational study in which the authors included a total of 17.740 patients, who underwent to coronary surgical revascularization between 2007 and 2016 and survived immediately and beyond 4 months after surgery, to analyse the impact of different BMI to long term outcomes, including all-cause mortality and major adverse cardio-cerebrovascular events (MACCEs). They divided the population into six groups based on their baseline BMI. The univariate analysis showed significantly higher all-cause mortality rates in the patients with BMI levels less than 18.5, between 18.5 and 25, and greater than 40 than in those with pre-obesity. After adjustments for several potential confounders, the analysis showed that the patients with BMI higher than 30 kg/m2 had a significantly higher risk of all-cause mortality than the pre-obesity group and a significant association was observed between the degree of obesity and all-cause mortality. Furthermore, the risk of 5-year MACCEs was significantly higher in the patients with BMI levels less than 18.5, between 18.5 and 24.9, and between 30 and 34.9 than in the pre-obesity group. The risk of MACCEs between all the groups with BMI greater than 35 kg/m2 and the pre-obesity group was similar. After adjustments for the potential confounders, a significant association was observed between the degree of obesity and the risk of 5-year MACCEs.

The authors concluded that the patients with obesity (BMI > 30 kg/m2) are at an increased risk of 5-year all-cause mortality and 5-year MACCEs and there is a significant positive association between the degree of obesity and the 5-year risks of all-cause mortality and MACCEs.

The topic of this study is very interesting and the potentialities of the analysis, including a large cohort of patients, are higher.

However, there are some points of discussion:

1. The English is acceptable, but could be improved.

2. The number of patients included in the analysis is specified in the section “Population” of the Results (lines 129-130). This information should be moved in the section “Study population” of the Material and Methods.

3. The Table 1 showed the baseline characteristics of the study population, including the preoperative risk factors and some surgical information. I suggest to divide the Table in two parts, “Preoperative characteristics” and “Intraoperative characteristics”, in order to make the table clearer and tidier.

4. The intraoperative characteristics could be implemented with additional data, such as the cardiopulmonary bypass time or the types of graft used for the coronary revascularizations.

5. At the line 196, “left main” is repeated.

6. In the analysis was not included the postoperative complications. Since the endpoints of the study were the long-term all-cause mortality and the major adverse cardio-cerebrovascular events (MACCEs), I think that is important to evaluate the incidence and the types of postoperative complications, that could affect the long-term survival of the patients and could increase the risk of mortality and of MACCEs.

7. There are several errors with the numbers of the references in the “Discussion” section. For example, at the lines 225, 232, 235. Please correct it.

8. The authors reported the total number of follow-up events considered in the analysis in the section “Endpoints”. I suggest to add the events, and the percentages, that occurred in the different groups. Moreover, these numbers should be reported in a Table, in order to make the article more complete and clearer.

9. It may be interesting add the causes of death in each group. These could be showed in a different table.

6. PLOS authors have the option to publish the peer review history of their article (what does this mean?). If published, this will include your full peer review and any attached files.

Reviewer #1: No

Reviewer #2: **Yes: **Marianna Berardi

---

## [Author Response · Author response to Decision Letter 0]

23 Jul 2022

Dear respected editor and reviewers;

Thank you very much for providing us with the opportunity to strengthen our manuscript. We sincerely appreciate your valuable comments. Having carefully considered the comments and suggestions, we have addressed our manuscript's requested amendments and revisions as outlined below in an itemized, point-by-point manner. We genuinely hope these changes meet the approval criteria of the esteemed reviewers and the editorial board. 

• In summary, we excluded BMI<18.5 patients, and we added medications for adjustments in our models. Subsequently, effect sizes were slightly attenuated. Nonetheless, the results of the Cox regressions and survival curves remained the same ( for BMI>40 and MACCE, the significant effect turned into a near-significant effect, which we assume is because of its relatively small sample size compared to other groups). We also provided the forest plot of our adjusted results. Next, we ran the RCS analyses to fit all-cause mortality and MACCE on BMI, the findings of which chimed in with the Cox regression results. The HR plot for the spline of BMI revealed a non-linear relationship between BMI and study outcomes, nevertheless to a negligible extent. We created the group "metabolic syndrome" as well, according to your requested classification. Analyses of metabolic syndrome revealed that the interaction term between diabetes, hyperlipidemia, and obesity was not sizable especially for mortality (near significant effects, however, for MACCE). These findings Implicated that the synergistic effects between these variables are,in fact, infinitesimal and negligible.

Answer: Many thanks for your guidance. All changes in format type are addressed and highlighted, including defining heading levels and the proper format for figures and tables, as well as references.

"The authors received no financial support for the research, authorship, and/or publication of this article. "

Answer: This study has been performed by resources of Tehran Heart Center and Cardiovascular Diseases Research Institute (our institutional budget and material). 

b) State what role the funders took in the study. If the funders had no role in your study, please state: "The funders had no role in study design, data collection and analysis, decision to publish, or preparation of the manuscript."

• Answer: The funders had no role in study design, data collection and analysis, decision to publish, or preparation of the manuscript

• Answer: None. 

d) If you did not receive any funding for this study, please state: "The authors received no specific funding for this work."

• Answer: The authors received no specific funding for this work

Answer: The amended statements are added to the cover letter. 

3. PLOS requires an ORCID iD for the corresponding author in Editorial Manager on papers submitted after December 6th, 2016. Please ensure that you have an ORCID iD and that it is validated in Editorial Manager. To do this, go to 'Update my Information' (in the upper left-hand corner of the main menu), and click on the Fetch/Validate link next to the ORCID field. This will take you to the ORCID site and allow you to create a new iD or authenticate a pre-existing iD in Editorial Manager. Please see the following video for instructions on linking an ORCID iD to your Editorial Manager account: https://www.youtube.com/watch?v=_xcclfuvtxQ

Answer: Kindly, we addressed your amendment in the new submission. 

4. Please include your full ethics statement in the 'Methods' section of your manuscript file. In your statement, please include the full name of the IRB or ethics committee who approved or waived your study, as well as whether or not you obtained informed written or verbal consent. If consent was waived for your study, please include this information in your statement as well.

Answer: The ethical statements are declared comprehensively in the method section. 

This is a retrospective study that studies the association between pre-operative BMI and 5-year outcomes after CABG.

I have reviewed the study too. Along with the other reviewers, please see and consider my comments below –

Answer: Dear editor, sincere thanks for your valuable comments. We are sincerely grateful for your time and consideration. We read your comments meticulously and tried to address those comprehensively. 

1. Please expand on the introduction and state what you aim to add in your study. Some actual results from prior studies would be good to report.

Answer: Sincere thanks for your comment. In order to expand the introduction in a brief yet comprehensive way, we included tangible results from three mata-analyses on this subject. In addition, the aims of the study are stated at the end of the introduction section. 

2. There are many issues in the methods section. Authors need to provide very clear definitions.

Please explain definitions for other variables in the study. DM - only type 2 or all, what were the baseline medications prior to surgery, please state and rephrase your observed events as primary and secondary endpoints, why opium use, is it very prevalent in your area?

Answer: Many thanks for your comments. We addressed them below. 

Variable definition adhered to the STS ( the Society of Thoracic Surgery) /SCA (the Society of Cardiovascular Anesthesiologists) guidelines ( as provided below) and is added to the method section

(The Society of Thoracic Surgeons (STS)/ the Society of Cardiovascular Anesthesiologists (SCA). STS SCA Data Specifications v2.9 Updated August 2019 [Available from: https://www.sts.org/sites/default/files/content/ACSD_Training%20Manual_V2-9%20Aug2019.pdf)

Hypertension was defined as a current diagnosis of hypertension defined by any 1 of the following: 

● History of hypertension diagnosed and treated with medication, diet, and/or exercise 

● Prior documentation of blood pressure >140 mm Hg systolic and/or 90 mm Hg diastolic for patients without diabetes or chronic kidney disease, or prior documentation of blood pressure >130 mm Hg systolic or 80 mm Hg diastolic on at least 2 occasions for patients with diabetes or chronic kidney disease 

● Currently undergoing pharmacological therapy for the treatment of hypertension. 

Diabetes was defined as : History of diabetes diagnosed and/or treated by a healthcare provider. The American Diabetes Association criteria include documentation of the following:

1. Hemoglobin A1c >=6.5%; or

2. Fasting plasma glucose >=126 mg/dL (7.0 mmol/L); or

3. 2-h Plasma glucose >=200 mg/dL (11.1 mmol/L) during an oral glucose tolerance test; or

4. In a patient with classic symptoms of hyperglycemia or hyperglycemic crisis, a random plasma glucose >=200 mg/dL (11.1 mmol/L)

Our definition includes patients with Type I DM but did not include gestational diabetes or steroid induced hyperglycemia. Also, patients were not categorized as diabetics merely based on consuming anti-diabetic agents because some medications used to treat diabetes may be used to treat other conditions. 

Hyperlipidemia

Hyperlipidemia was defined if the patient had a history of hyperlipidemia that was diagnosed and/or treated by a physician. Also, it was defined based on NCEP criteria including

documentation of the following:

● Total cholesterol >200 mg/dL (5.18 mmol/L); or

● LDL >=130 mg/dL (3.37 mmol/L);

● Currently receiving antilipidemic treatment

Current cigarette smoking was defined as consuming at least 100 cigarettes in total in a person who has been smoking for at least one previous month. 

Opium consumption was defined as current or former use of opium through ingestion or smoking. 

 Sincerely, Regarding the prevalence of opium consumption that you addressed, the prevalence is considerable among the Iranian general population (5-17% in different population-based studies). There is strong body of evidence that opium consumption is a risk factor for CVD (references: Opium consumption and coronary atherosclerosis in diabetic patients: a propensity score-matched study (Planta Medica) / Effects of opium consumption on cardiometabolic diseases (Nature Reviews Cardiology)/ Opium and cardiovascular health: A devil or an angel? ( Indian Heart Journal) / Does Opium Consumption Have Shared Impact on Atherosclerotic Cardiovascular Disease and Cancer? (Archives of Iranian Medicine)). The largest Iranian cohort study on 50,000 individuals – the Golestan Cohort study- reported that the risk of death from ischemic heart disease in opium users was 1.9 times that in non-users (reference: Opium use and mortality in Golestan Cohort Study: prospective cohort study of 50 000 adults in Iran (British Medical Journal)). Our recent cohort survey of 28,961 patients who underwent CABG at our center revealed that persistent opium consumption after CABG was a significant independent predictor of increased 5-year mortality (HR: 1.28, P value:0.009 ) and MACCE (major adverse cardio-cerebrovascular events (HR:1.25, P-value <0.001 ) (reference: Effect of persistent opium consumption after surgery on the long-term outcomes of surgical revascularization (European Journal of Preventive Cardiology)). Hence, we decided to include opium consumption as an essential confounder when conducting our analyses. 

Positive family history of coronary artery disease was defined as having any of the following events among first-degree females relatives <65 years and first-degree male relatives <55 years old: sudden death / myocardial infarction/ PCI/ CABG/ positive coronary angiography ( stenosis >50 % in at least one coronary artery). 

CVA/TIA was obtained by either patient's past medical history or by a neurologist consult based on the patient's situation and/or brain imaging. Stroke was defined as an acute episode of focal or global neurological dysfunction caused by brain, spinal cord, or retinal vascular injury resulting from hemorrhage or infarction, where the neurological dysfunction lasts for more than 24 hours. Transient ischemic attack (TIA) was defined as a transient episode of focal neurological dysfunction caused by the brain, spinal cord, or retinal ischemia, without acute infarction, where the neurological dysfunction resolves within 24 hours

COPD was defined based on the patient's medical history or on spirometry results ( mostly FEV1/FVC and FEV1% of predicted), indicating irreversible obstruction of airways. 

FEV1 > 75% of predicted = Normal

FEV1 60% to 75% of predicted = Mild obstruction

FEV1 50% to 59% of predicted = Moderate obstruction

FEV1 < 50% of predicted = Severe obstruction

Renal failure was excluded from table 1, and CKD ( with the definition of eGFR<60 as you asked in comment #10) was replaced instead.

Prolonged ventilation is newly added due to comment #6 from respected reviewer #2 and was defined as greater than 24 hours of ventilation following surgery.

Regarding primary and secondary endpoints, in our study, all-cause mortality and MACCE were the primary outcomes, and we did not specify secondary outcomes. We add this statement to the revised manuscript under the "study endpoints" section. 

Regarding patients’ medications, since discharge medications have more considerable impact on patients’ post-operative outcomes than pre-operative medications, we included those in our analyses. Selected medications were those known to affect CABG patients' survival, including beta-blockers, statins, aspirin/other anti-platelets, and ACEI/ARBs (Angiotensin-converting enzyme inhibitor/ Angiotensin II receptor blockers. 

3. For table 1, please provide an overview of all patients and then according to category of BMI. BMI groups are according to the WHO. Please state that in the methods section. It would also be good to report BMI and a continuous variable and maybe provide a histogram of distribution.

Answer : We added the statement about BMI categorization in the revised manuscript under the section "Body Mass Index Classification". The distribution of BMI by histogram is added in supplementary material, cited within the text in the result section, as is also presented below. An overview of the study cohort is integrated into table 1 in the revised manuscript. Also, continuous BMI is reported in Table 1 in the new version. Since in your comment #4 you advised us to omit the group BMI<18.5 ( which we did) the histogram is provided for the new study population (n=17751) without this group. 

4. What was your median and maximum follow-up period ? Both need to be reported. Please report the event rates per category first and then further analyses. Results may not be very robust as there are only 141 patients in the < 18.5 group. I would recommend combining < 18.5 & 18.5 – 25 groups. For your study, I would in fact remove < 18.5. You have very few patients in this group and they may be frail people, or very fit people which are both different category of patients.

Answer: As you stated we excluded patients with BMI<18.5 from all of the analyses ,and we stated the reason in the method section. Accordingly, the number of the study population and numbers in the tables are revised (Tables 1 ,2,3). Accordingly, respective figures are depicted with new results.

The new ( with BMI<18.5 omitted) median and `maximum follow-up time are reported in the revised version under the subsection "follow-up" of the "result" section. The new figures are as follows. The median follow-up was 60.1 [95%CI : 59.2-60.9] months. The maximum follow-up was 133.8 months.149 patients were lost to follow-up. Event rate per category is added to the subsection "Endpoint" of the "results" section. Some findings are added and reported in the revised manuscript, and some are cited in the supplementary table, which is also provided below for your convenience. 

 All patients

N=17751 18.5≤BMI<25

n= 5547 25≤BMI<30

n= 8091 30≤BMI<35

n= 3304 35≤BMI<40

n=661 BMI≥40

n=148

All-cause mortality 1838

(10.4) 653

(11.8) 783

(9.7) 313

(9.5) 66

(10.0) 23

(15.5)

MACCE ( first-event) 3540

(19.9) 1163

(21.1) 1547

(19.3) 667

(20.2) 132

(20.3) 31

(21.1)

MACCE components 

(first-event)

 ACS 1471

(8.3) 433

(7.9) 658

(8.1) 304

(9.2) 62

(9.6) 10

(6.8)

 CVA 412

(2.3) 140

(2.5) 181

(2.3) 80

(2.4) 11

(1.7) 0

 Death 1661

(9.4) 590

(10.7) 708

(8.8) 283

(8.6) 59

(9) 21

(14.3)

5. Please state clearly all the variables used for adjustment in the model. I would recommend that authors use a spline term to fit BMI and also present results of that regression model.

Answer: Sincerely, all the variables that were adjusted for were previously stated in the "Endpoint" subsection of the results and in the footnote of Table 2 and 3. We moved the variables in the manuscript to the method section, and they remained in the tables' footnotes. Due to the comments, there were some changes in the variables, which are noted in the revised manuscript. These included 1. Changing GFR to CKD (according to your comment #10) 2. Integrating discharge medications known to affect CABG patients' survival into the analyses. These included beta-blockers, statins, aspirin/other anti-platelets, and ACEI/ARBs (angiotensin-converting enzyme inhibitor/ Angiotensin II Receptor Blockers. Kindly, regarding other variables that you mentioned (opium , ICU stay, and PAD) we tried to explain in their respective comments why we could not include/exclude those too. Should any other changes be needed, please let us know. 

As your requested, we applied the Restricted Cubic Splines (RCS) models for all-cause mortality and MACCEs to fit BMI. Variables used for adjustments were the same as in the Cox regression models (age, gender, diabetes mellitus, hypertension, hyperlipidemia, positive family history, current smoking, opium abuse, CKD, ejection fraction, left main involvement, number of diseased vessels, number of grafts, ICU stay, off-pump surgery, recent MI ( MI under 7 days) COPD, CVA/TIA, previous PCI, and discharge medications ( ACEI/ARB- beta blockers- statins- aspirin/other antiplatelet) . We applied five knots (df=4) with a hazard scale. The results of RCS models chimed in with those of Cox regression. As can be seen in the tables below, continuous BMI had a significant positive association with all-cause mortality (adjusted HR: 1.02 , 95% CI: 1.02-1.05 , P value <0.001) and MACCE (adjusted HR: 1.02, 95% CI: 1.01-1.03, P value <0.001). Therefore, our observations in the Cox regression models and RSC models indicated a significant association between the degree of obesity and all-cause mortality and MACCE at the median follow-up of 5 years. We Included the results of RCS analyses in the result section and supplementary materials. The RCS analyses are also added to the statistical analyses section. 

Restricted Cubic Spline for BMI and all-cause mortality 

Variable HR 95%CI P value 

BMI 1.03 1.02-1.05 <0.001

Age 1.05 1.05-1.06 <0.001

Male 2.51 2.16-2.93 <0.001

Diabetes 1.45 1.31-1.61 <0.001

Hypertension 1.35 1.21-1.51 <0.001

Hyperlipidemia 0.91 0.81-1.02 0.108

Positive family history 0.99 0.88-1.1 0.791

Current smoking 1.21 1.05-1.4 0.011

Opium 1.21 1.04-1.4 0.013

CKD 1.74 1.54-1.96 <0.001

EF 0.97 0.96-0.97 <0.001

Left main 0.98 0.83-1.16 0.794

Diseased vessels 1.26 1.12-1.41 <0.001

Graft number 0.89 0.84-0.95 0.001

ICU Hours 1.001 1.001-1.001 <0.001

Off Pump 1.07 0.87-1.33 0.5

Recent MI 0.99 0.83-1.18 0.903

COPD 1.3 1.04-1.63 0.024

CVA/TIA 1.53 1.3-1.8 <0.001

Previous PCI 0.96 0.74-1.25 0.761

ACEI/ARB 0.96 0.87-1.07 0.484

ASA/anti-platelets 0.39 0.32-0.46 <0.001

Statins 0.47 0.4-0.55 <0.001

Beta blockers 0.56 0.48-0.65 <0.001

RCS1 1.98 1.92-2.04 <0.001

RCS 2 0.87 0.85-0.89 <0.001

RCS 3 0.86 0.84-0.87 <0.001

RCS 4 0.98 0.96-1 0.013

RCS 5 0.99 0.97-1 0.029

Constant 0.01 0-0.02 <0.001

HR, Hazard ratio; CI,Confidence interval; BMI, Body mass index; CKD, Chronic kidney disease; EF, Ejection fraction; SVD,Single vessel disease; VD, Vessel disease; ICU, Intensive care unit; MI, Myocardial infarction; COPD, Chronic obstructive pulmonary disease; CVA, Cerebrovascular accidents; TIA, Transient ischemic attack; PCI, Percutaneous coronary intervention; ACIE, Angiotensin converting enzyme inhibitor,; ARB, Angiotensin II receptor blocker; ASA, Aspirin 

Rescrticted Cubic Spline for BMI and MACCE

Variable HR 95% CI P value 

BMI 1.02 1.01-1.03 <0.001

Age 1.02 1.01-1.02 <0.001

Male 1.27 1.16-1.39 <0.001

Diabetes 1.29 1.2-1.39 <0.001

Hypertension 1.28 1.19-1.39 <0.001

Hyperlipidemia 0.92 0.85-0.99 0.033

Positive family history 1.02 0.94-1.1 0.67

Current smoking 1.1 0.99-1.22 0.08

Opium 1.11 0.99-1.23 0.066

CKD 1.45 1.32-1.59 <0.001

EF 0.98 0.98-0.99 <0.001

Left main 0.96 0.85-1.09 0.563

Disease vessels 1.09 1.01-1.18 0.029

Graft number 0.92 0.88-0.97 0.001

ICU Hours 1.001 1.001-1.001 <0.001

Off Pump 1 0.86-1.17 0.973

Recent MI 1.01 0.89-1.15 0.839

COPD 1.16 0.97-1.38 0.094

CVA/TIA 1.4 1.24-1.59 <0.001

Previous PCI 1.26 1.07-1.48 0.006

ACEI/ARB 1.01 0.94-1.09 0.777

ASA/anti-platelets 0.5 0.43-0.58 <0.001

Statins 0.61 0.54-0.7 <0.001

Beta blockers 0.7 0.62-0.79 <0.001

Rcs1 2.14 2.09-2.2 <0.001

Rcs2 0.94 0.91-0.96 <0.001

Rcs3 0.88 0.87-0.9 <0.001

Rcs4 1 0.98-1.01 0.48

Rcs5 1 0.99-1.01 0.908

Constant 0.19 0.11-0.32 <0.001

MACCE, Major cardio-cerebrovascular events; HR, Hazard ratio; CI,Confidence interval; BMI, Body mass index; CKD, Chronic kidney disease; EF, Ejection fraction; SVD,Single vessel disease; VD, Vessel disease; ICU, Intensive care unit; MI, Myocardial infarction; COPD, Chronic obstructive pulmonary disease; CVA, Cerebrovascular accidents; TIA, Transient ischemic attack; PCI, Percutaneous coronary intervention; ACIE, Angiotensin converting enzyme inhibitor,; ARB, Angiotensin II receptor blocker; ASA, Aspirin 

6. Please provide adjusted analyses results in a separate table, or a forest plot would be better.

Answer: Many thanks for your advice. We provided a forest plot as requested and included that in the revised manuscript. You may also see the plots below, depicting worsening outcomes with BMI increasing. 

7. Figures – please provide a simple cumulative plot for all-cause mortality and MACCE for the whole group with confidence intervals. Then provide figures for each BMI category. I would prepare separate plots for each BMI category and provide confidence intervals and # patients at risk for each time point listed on the x axis. It would be good to also see a HR plot for the spline of BMI as a continuous variable to see if the increase in HR is non-linear for increasing BMI.

• Answer: Many thanks for your noteworthy comment. As you asked, the simple cumulative plot for our outcomes for the whole cohort is presented below and is also added to the revised manuscript. 

• Upon your request, we also provided separate (unadjusted) plots for each BMI category with the number of patients at risk and CIs demonstrated below. 

• Appreciatively, since we assumed that these figures are to serve as a means to draw comparisons between BMI groups, we think that providing information in separate figures might not fulfill this purpose. Therefore, we also provided unadjusted plots for mortality and MACCE in which all BMI groups are presented with respective HRs and the number of at-risk patients. However, due to confidence intervals getting mixed up in the plots and thereby being non-informative, we omitted CIs. If you will, we would prefer to include these figures (rather than separate plots) in the manuscript for now) we integrated them into Figure 3 and 4 of the revised MS and changed figure legends accordingly). Kindly, if from your professional perspective, the separate plots can convey the concept better, we will do so at the next step. Please find the figures and our interpretations below. 

• The unadjusted figures confirmed the Cox model findings on hazard ratios at the univariable level. It was observed that the unadjusted mortality risk was higher by a wide margin among patients with BMI≥40 compared to the pre-obesity group. In addition, a slightly higher mortality risk was observed among 35≤BMI<40 (especially during the first half of the study period) and 18.5≤BMI<25 (especially during the second half of the study period) compared to the referent. As for the MACCE composite, it was observed that the risks in different BMI categories are closer together, however the unadjusted risks were slightly higher among all BMI groups compared to the referent (25≤BMI<30). 

• We also provided an HR plot for the spline of BMI to evaluate the linearity of the association between BMI and outcomes. Please see the figure and our interpretations below. We included these results in the revised manuscript as well. 

• As can be seen for both mortality and MACCE, albeit changes in HR are to some extent variable, considering the HR scale axis, these variations are not considerable ( ≈≤2%), and therefore we should not be concerned about the previous results on categorical BMI. Especially that, the points at which the curve's slope changes are almost compatible with BMI cut-off points. Finally, it is true that the association between BMI and outcomes is, to some extent, non-linear; but this non-linearity seems negligible given the small HR variation ( we scaled the graphs so that we could detect any non-linearity. In other words, we observed that the BMI effects of not fixed, however, its variations are not sizable (by up to ≈2% in mortality and up to ≈1% in MACCE) ; thus, we belive that reporting the effect sizes of BMI in categories does not really over/under estimate its effects on our outcomes. 

8. As reviewer states, the title provides a causal link, but this is a paper looking at association not causation. – change the title please to remove this causal language.

Answer: Thank you for your notice. We changed the title to "The Association Between Different Body Mass Index Levels and Long-term Surgical Revascularization Outcome."

9. In the abstract and paper, you need to clearly state that this is 'pre-operative BMI' and again provide BMI as a continuous variable before splitting it into groups.

Answer: Appreciatively, the term "pre-operative was added to both the abstract and the method section (study population) of the manuscript to clarify this concept. Moreover, the figures for continuous BMI are stated in the abstract and the results (study population) in the revised manuscript. 

10. There is no mention regarding medications that patients are on. These should be used to adjust for in the model. Some variables like ICU stay, opium use, are not very meaningful for 5-year outcomes and can be removed from the model. Rather than graft # and number of diseased vessels, complete vs incomplete revascularization would be better. MI under 7 hours can be changed to recent MI. eGFR can be changed to CKD with CKD – eGFR < 60. That would be more clinically meaningful.

Answer: 

• Many thanks for your valuable notice. Unfortunately, our databank does not provide us with the results of "complete vs. incomplete revascularization". Therefore, we used graft numbers and numbers of diseased vessels as variables at our disposal. MI under 7 days was renamed to recent MI and defined in the method section. eGFR was changed to CKD, as you stated. Also, CKD was used for the adjustments rather than eGFR for new analyses. 

We tried to mention the importance and prevalence of opium consumption in Iran, especially among patients with cardiovascular diseases, earlier in response to your comment #2. That was why we decided to include this variable in the model. ICU stay was chosen in our model as a surrogate index (rather than a confounder) reflecting the overall status of the patients. In our opinion, therefore, it was a valuable marker presenting the occurrence of many known and unknown post-operative events and confounders which might affect long-term survival, such as acute kidney injury, infections, arrhythmias and so on. As respected reviewer #2 asked us in comment #6 about the importance of post-operative complications, we chose ICU hours as the representative of all known and unknown confounders/complications after surgery. Choosing ICU hours instead of individual post-operative complications in the model allowed us to present the model as brief yet punctual as possible. As another justification, we could not obtain data on all possible kinds of operative complications from our databank. Previous studies have reported prolonged ICU stay as an independent predictor of long-term survival. In this regard, Balakrishnan Mahesh et al. revealed that among 6,101 patients who underwent cardiac surgery, the 3-year survival of patients with prolonged ICU stay (more than 72h) was 81.2% compared to 93.6% in the control group without prolonged stay ( P value <0.001) (reference: Prolonged stay in intensive care unit is a powerful predictor of adverse outcomes after cardiac operations (The Annals of thoracic surgery). This finding has been replicated in another study on CABG patients (median follow-up: 31 months) who showed that patients who required a prolonged ICU stay (more than 48h) had significantly lower survival and freedom from cardiac readmission to the hospital. Prolonged ICU stay was an independent predictor of the composite outcome of "death and readmission" (HR: 1.8 (1.5-2.1) (reference: Long-term outcomes in patients requiring stay of more than 48 hours in the intensive care unit following coronary bypass surgery (Journal of critical care)).As a consequence, in order to reduce the confounding effect of the ICU stay on outcomes we integrated it into the model. Appreciatively, still, if respected reviewers and the respected editor consider the removal of ICU stay and instead adding post-op complications (added to Table 1.) in the analytic models beneficial, we will do so in the next step. 

11.Rather than only considering BMI, can authors also combine BMI, DM, and hyperlipidemia to identify those with metabolic syndrome and also present results for patients with and without metabolic syndrome.

Answer: Dear esteemed editor, herein as you asked, we combined the following groups, and we named that as "metabolic syndrome group": BMI>30, positive diabetes mellitus, positive hyperlipidemia. Below are the results and interpretations of the analyses on all-cause mortality and MACCE. Respectfully, at this time, we feel that the mentioned results are out of the scope of this manuscript's purpose. If you will, therefore, we have not yet added these findings to the manuscript. We are, however, eager to present these findings in a separate original article or a letter with a more related topic in which we can also provide more detailed analyses if the respected editorial board wishes so. Otherwise, we will add these new results to the next version on your demand. 

Metabolic syndrome and all-cause mortality 

 HR 95% CI P value 

UNADJUSTED

Metabolic syndrome 1.02 0.82-1.27 0.873

INTERACTION

Metabolic syndrome 1.02 0.78-1.33 0.878

Hyperlipidemia 0.73 0.65-0.82 <0.001

Diabetes mellitus 1.53 1.39-1.68 <0.001

BMI 

[18.5, 25) 1.21 1.09-1.35 <0.001

[30, 35) 1.01 0.88-1.16 0.909

[35, 40) 1.08 0.83-1.39 0.568

>=40 1.83 1.2-2.78 0.005

ADJUSTED

Metabolic syndrome 1.05 0.79-1.39 0.74

Hyperlipidemia 0.9 0.8-1.02 0.114

Diabetes mellitus 1.45 1.31-1.62 <0.001

BMI

[18.5, 25) 0.91 0.81-1.03 0.124

[30, 35) 1.18 1.01-1.38 0.034

[35, 40) 1.74 1.31-2.32 <0.001

>=40 2.9 1.82-4.62 <0.001

Age 1.05 1.05-1.06 <0.001

Male 2.56 2.19-2.98 <0.001

Hypertension 1.36 1.22-1.51 <0.001

Family history 0.99 0.89-1.1 0.831

Current smoking 1.21 1.05-1.4 0.01

Opium 1.22 1.05-1.41 0.01

CKD 1.72 1.52-1.94 <0.001

EF 0.97 0.96-0.97 <0.001

Left main 0.97 0.82-1.15 0.731

Vessel Disease

2VD 0.82 0.59-1.15 0.248

3VD 1.13 0.82-1.57 0.461

Graft number 0.9 0.84-0.96 0.001

ICU Hours 1.001 1.001-1.001 <0.001

Off Pump 1.07 0.87-1.32 0.536

Recent MI 0.99 0.83-1.17 0.877

COPD 1.28 1.02-1.61 0.031

CVA/TIA 1.53 1.3-1.8 <0.001

Previous PCI 0.96 0.74-1.25 0.757

Discharge medications 

ACEI/ARB 0.96 0.87-1.07 0.48

ASA/anti-platelets 0.39 0.32-0.47 <0.001

Beta blockers 0.56 0.49-0.65 <0.001

Statins 0.46 0.39-0.54 <0.001

Metabolic syndrome and MACCE

 HR 95% CI P value 

UNADJUSTED

Metabolic syndrome 1.25 1.08-1.44 0.003

INTERACTION

Metabolic syndrome 1.13 0.94-1.35 0.187

Hyperlipidemia 0.84 0.78-0.91 <0.001

Diabetes mellitus 1.38 1.29-1.48 <0.001

BMI 

[18.5, 25) 1.11 1.02-1.19 0.01

[30, 35) 1.08 0.98-1.19 0.142

[35, 40) 1.09 0.91-1.31 0.368

>=40 1.18 0.83-1.69 0.361

ADJUSTED

Metabolic syndrome 1.16 0.97-1.4 0.109

Hyperlipidemia 0.9 0.83-0.98 0.014

Diabetes mellitus 1.27 1.18-1.37 <0.001

BMI

[18.5, 25) 0.97 0.9-1.06 0.516

[30, 35) 1.11 1-1.24 0.054

[35, 40) 1.23 1.01-1.5 0.041

>=40 1.27 0.85-1.88 0.241

Family history 1.02 0.94-1.09 0.673

Current smoking 1.09 0.99-1.21 0.089

Opium 1.11 0.99-1.23 0.068

CKD 1.43 1.31-1.57 <0.001

EF 0.98 0.98-0.98 <0.001

Left main 0.96 0.85-1.09 0.542

Vessel Disease

2VD 0.87 0.7-1.08 0.212

3VD 1.01 0.82-1.24 0.936

Graft number 0.93 0.89-0.97 0.001

ICU Hours 1.001 1.001-1.001 <0.001

Off Pump 1 0.86-1.16 0.976

Recent MI 1.01 0.9-1.15 0.828

COPD 1.15 0.97-1.37 0.11

CVA/TIA 1.4 1.24-1.59 <0.001

Previous PCI 1.26 1.07-1.48 0.006

ACEI/ARB 1.01 0.94-1.09 0.802

ASA/anti-platelets 0.5 0.43-0.58 <0.001

Beta blockers 0.7 0.62-0.78 <0.001

Statins 0.61 0.54-0.7 <0.001

We assume that evaluating the three mentioned comorbidities together is actually a kind of interaction term between these variables. Therefore, in the interaction model, we will refer to "metabolic syndrome" (MS) as the interaction term. For all-cause mortality, MS did not exert any significant effect, nor did its effect size and P value change considerably in the interaction or adjusted model (in which we included three main variables along with other covariates and MS as an interaction term). Therefore, no synergistic effects between obesity, diabetes, and hyperlipidemia was observed for mortality. In the unadjusted model for the MACCE composite, MS significnalty increased the risk. Subsequently, in the interaction and adjusted models were obsereved a borderline effect (near-significant P values). Therefore, there might have been an effect which we could not track down. Nevertheless, given the effects sizes, the interaction did not really exist (if it did, after adding diabetes and hyperlipidemia, the effects sizes would surge in the adjusted and interaction models). 

12. presence of PAD is very important as a risk factor and should be reported in table 1 and included in the Cox model.

Answer: Genuine appreciation for your notice. Unfortunately, missing data on peripheral vascular diseases is high in our data bank, and it is, therefore, not reliable. That was the reason we did not include it in the models. In the revised version, we addressed this limitation in the "study strengths and limitations" subsection. 

13. I would not consider 5 years to be long term for CABG outcomes; long term for CABG would be 10 years and beyond. Please change long term to mid-term.

Answer: Your requested change is addressed in the revised manuscript and the term "long-term" is changed to "mid-term"

14. Please restructure the discussion as follows –

P1 = what we have observed

P2 = current literature and how what we have found is the same or different / why if different ?

P3 = clinical implications of our findings 

P4 = Strength and limitations.

Answer: The discussion is restructured as you stated, and its parts are divided into subsections. Some statements are added in the revised manuscript. 

15. Data on follow-up was collected by visits. Do you have BMI at follow up and can you model change in BMI and outcomes? Most papers only look at pre-operative BMI and change in BMI would be very interesting to see.

Answer: We do believe that it would be of great importance and interest if we could integrate follow-up BMI into our analytic models. Regrettably, it is not possible for us because the weight was not included in our CABG follow-up forms. We have addressed this hurdle in our limitations.

Review Comments to the Author

Reviewer #1: 

This study does represent an interesting topic in obesity paradox. This study does have clinical priority however there exist many ways in which an unobserved covariate or several various factor lead to confounding to explain results. As discussed in study, BMI was not serially monitored in the patients in follow-up given the categories the patients were stratified to which may inappropriately bias them into a cohort. I believe the title is misleading as their is no effect of BMI on revascularization outcomes but found associations in particular to obese individuals.

Answer: Dear valued reviewer, many thanks for your interest, consideration, and comments. As you have mentioned, albeit reasonably markedly important; unfortunately, we are unable to add post-operative BMI values in our analyses. We tried to address this obstacle in the study limitations, and we also revised the title accordingly based on your comment. Due to this kind of issue, our colleagues and we have integrated many new variables in the follow-up forms, one of which is weight, and therefore we hope that we can evaluate the implications of follow-up BMI on outcomes in the not-too-distant future. Regarding the residual confounding effects, we would like to assure you that we tried to adjust for as many confounders as possible according to the capacity and characteristics of our databank. In the new analyses are also included discharge medications. Nevertheless, we acknowledge that the confounding effects might still persist with other known and unknown variables, especially those related to follow-up information. 

Reviewer #2: 

This is a retrospective observational study in which the authors included a total of 17.740 patients, who underwent to coronary surgical revascularization between 2007 and 2016 and survived immediately and beyond 4 months after surgery, to analyses the impact of different BMI to long term outcomes, including all-cause mortality and major adverse cardio-cerebrovascular events (MACCEs). They divided the population into six groups based on their baseline BMI. The univariate analysis showed significantly higher all-cause mortality rates in the patients with BMI levels less than 18.5, between 18.5 and 25, and greater than 40 than in those with pre-obesity. After adjustments for several potential confounders, the analysis showed that the patients with BMI higher than 30 kg/m2 had a significantly higher risk of all-cause mortality than the pre-obesity group and a significant association was observed between the degree of obesity and all-cause mortality. Furthermore, the risk of 5-year MACCEs was significantly higher in the patients with BMI levels less than 18.5, between 18.5 and 24.9, and between 30 and 34.9 than in the pre-obesity group. The risk of MACCEs between all the groups with BMI greater than 35 kg/m2 and the pre-obesity group was similar. After adjustments for the potential confounders, a significant association was observed between the degree of obesity and the risk of 5-year MACCEs.

The authors concluded that the patients with obesity (BMI > 30 kg/m2) are at an increased risk of 5-year all-cause mortality and 5-year MACCEs and there is a significant positive association between the degree of obesity and the 5-year risks of all-cause mortality and MACCEs.

The topic of this study is very interesting and the potentialities of the analysis, including a large cohort of patients, are higher.

However, there are some points of discussion:

Answer: Dear appreciated reviewer, please accept our sincere thanks for your consideration and comments aimed at improving our work. Please find our point-by-point responses below. 

1. The English is acceptable, but could be improved.

Answer: Thank you for your notice. In the revised veriosn, we had our manuscript edited by a professional language editor ,with more than 17 years of expertise, who have worked with Tehran Heart Center (THC) journlas and many others. We hope that the edit meets your language standards. 

2. The number of patients included in the analysis is specified in the section "Population" of the Results (lines 129-130). This information should be moved in the section "Study population" of the Material and Methods.

Answer: Sincerely, your requested change is addressed in the revised manuscript. 

3. Table 1 showed the baseline characteristics of the study population, including the pre-operative risk factors and some surgical information. I suggest to divide the Table in two parts, "Pre-operative characteristics" and "Intraoperative characteristics", in order to make the table clearer and tidier.

Answer: Sincerely, your requested changes is addressed in the revised manuscript and the patient's characteristics are categorized into pre- and post-operative details. Furthermore, some new variables are added. 

4. The intraoperative characteristics could be implemented with additional data, such as the cardiopulmonary bypass time or the types of graft used for the coronary revascularizations.

Answer : Sincerely, data on the cardiopulmonary bypass (CPB) time, number of arterial and venous graft, use of internal mammary artery, urgent/emergent surgery , and peri-operative IABP are added to Table 1 of baseline characteristics.

5. At the line 196, "left main" is repeated.

Answer: It is corrected in the revised version in the foot notes of Table 2 and 3. 

6. In the analysis was not included the post-operative complications. Since the endpoints of the study were the long-term all-cause mortality and the major adverse cardio-cerebrovascular events (MACCEs), I think that is important to evaluate the incidence and the types of post-operative complications, that could affect the long-term survival of the patients and could increase the risk of mortality and of MACCEs.

Answer: Dear esteemed reviewer, thank you so much for your notice. As you mentioned, we do believe that post-operative complications can affect mid/long-term consequences. Therefore, in the revised version is added information on many post-operative complications (Table 1). Among the study cohort, 28.4% had blood transfusion in the ICU, 0.8% experienced CVA/TIA, 2.1% had prolonged ventilation, and 2.4% underwent reoperation for tamponade or bleeding. As for including complications into analyses, we actually used "ICU stay" in the model as a surrogate index since it is increased in all of the abovementioned complications. ICU stay was chosen in our model as a surrogate index reflecting the overall status of the patients. In our opinion, therefore, it could reflect the occurrence of many known and unknown post-operative complications which might affect long-term survival, such as acute kidney injury, infections, arrhythmias, and so on. Choosing ICU hours instead of individual post-operative complications in the models allowed us to present the model as brief, yet punctual, as possible. As another justification, we could not obtain data on all possible kinds of operative complications from our databank. Appreciatively, still, if you consider using individual complications rather than ICU hours, we will do so in the next step. 

7. There are several errors with the numbers of the references in the "Discussion" section. For example, at the lines 225, 232, 235. Please correct it.

Answer : Thank you for your attention. We put all the reference numbers before dots and commas in the revised manuscript. Sincerely, should there any other changes be needed, please let us know. 

8. The authors reported the total number of follow-up events considered in the analysis in the section "Endpoints". I suggest to add the events, and the percentages, that occurred in the different groups. Moreover, these numbers should be reported in a Table, in order to make the article more complete and clearer.

Answer: Thank you for your comment. The event rate per category is added to the subsection "Endpoint" of the "results" section. Some relevant findings are added and reported in the revised manuscript, and some are cited in the supplementary table. Please also find the event rates per category below. 

 All patients

N=17751 18.5≤BMI<25

n= 5547 25≤BMI<30

n= 8091 30≤BMI<35

n= 3304 35≤BMI<40

n=661 BMI≥40

n=148

All-cause mortality 1838

(10.4) 653

(11.8) 783

(9.7) 313

(9.5) 66

(10.0) 23

(15.5)

MACCE ( first-event) 3540

(19.9) 1163

(21.1) 1547

(19.3) 667

(20.2) 132

(20.3) 31

(21.1)

MACCE components 

(first-event)

 ACS 1471

(8.3) 433

(7.9) 658

(8.1) 304

(9.2) 62

(9.6) 10

(6.8)

 CVA 412

(2.3) 140

(2.5) 181

(2.3) 80

(2.4) 11

(1.7) 0

 Death 1661

(9.4) 590

(10.7) 708

(8.8) 283

(8.6) 59

(9) 21

(14.3)

BMI, body mass index; MACCE, major adverse cardio-cerebrovascular events; ACS, acute coronary syndrome; CVA, cerebrovascular events

Data are presented as number and frequency. 

9. It may be interesting add the causes of death in each group. These could be showed in a different table.

Answer: Thank you genuinely for your thought-provoking suggestion. We do collect the cause of death in our follow-up forms. Nevertheless, we do not possess a "cause of death ascertainment protocol", and that was why these data were not reported in our manuscript. In fact, information on the cause of death is based on families' self-reports, not autopsy results or medical documents of the patients. Also, the cause of death of 17% of the patients was missing (therefore, we reported a valid percent). These findings should therefore be interpreted with caution. The table below provides the cause of death. Due the possibily of it not being accurate, we have not reported these findings in the manuscript. 

Cause of death All patients 

N=17751 18.5≤BMI<25

n= 5547 25≤BMI<30

n= 8091 30≤BMI<35

n= 3304 35≤BMI<40

n=661 BMI≥40

n=148

Cardiovascular 686)45.0%) 256)45.6%) 282)43.5%) 117)42.6%) 19)42.4%) 12)63.2%)

Renal 71(4.7%) 28)5.0%) 25)3.9%) 14)5.5%) 3)6.7%) 1)5.3%)

Cancer 271)17.8%) 105)18.7%) 117)18.1%) 43)17.0%) 6)13.3%) 0

CVA 177)11.6%) 61)10.9%) 72)11.1%) 33)13.0%) 9)20%) 2)10.5%)

Other causes 247 (16.2%) 86 (15.3%) 115 (17.7%) 35 (13.8%) 7 (15.6%) 4 (21.1%)

Unknown 74 (4.8%) 25 (4.5%) 37 (5.7%) 11 (4.3%) 1 (2.2%) 0 

CVA, cerebrovascular accidents 

NewCauseDeath * BMI_cat Crosstabulationa

 BMI_cat Total

 [25, 30) [18.5, 25) [30, 35) [35, 40) >=40 

NewCauseDeath Cardiac Count 282 256 117 19 12 686

 % within NewCauseDeath 41.1% 37.3% 17.1% 2.8% 1.7% 100.0%

 % within BMI_cat 43.5% 45.6% 46.2% 42.2% 63.2% 45.0%

 Renal Count 25 28 14 3 1 71

 % within NewCauseDeath 35.2% 39.4% 19.7% 4.2% 1.4% 100.0%

 % within BMI_cat 3.9% 5.0% 5.5% 6.7% 5.3% 4.7%

 Cancer Count 117 105 43 6 0 271

 % within NewCauseDeath 43.2% 38.7% 15.9% 2.2% 0.0% 100.0%

 % within BMI_cat 18.1% 18.7% 17.0% 13.3% 0.0% 17.8%

 CVA Count 72 61 33 9 2 177

 % within NewCauseDeath 40.7% 34.5% 18.6% 5.1% 1.1% 100.0%

 % within BMI_cat 11.1% 10.9% 13.0% 20.0% 10.5% 11.6%

 other Count 115

17.7% 86

15.3% 35

13.8% 7

15.6% 4

21.1% 247

 % within NewCauseDeath 

 % within BMI_cat 16.2%

 Unknown Count 37

5.7% 0 74

 % within BMI_cat 0.0% 4.8%

Total Count 648 561 253 45 19 1526

 % within NewCauseDeath 42.5% 36.8% 16.6% 2.9% 1.2% 100.0%

 % within BMI_cat 100.0% 100.0% 100.0% 100.0% 100.0% 100.0%

a. Mortality_code = Yes

Best Regards, 

Jamshid Bagheri, MD (corresponding author)

Cardiovascular diseases research institute, 

Tehran University of Medical Sciences, 

Tehran, Iran

---

## [Decision Letter · Decision Letter 1]

23 Aug 2022

The Association Between Different Body Mass Index Levels and Midterm Surgical Revascularization Outcomes

PONE-D-22-09560R1

Dear Dr. Bagheri,

We’re pleased to inform you that your manuscript has been judged scientifically suitable for publication and will be formally accepted for publication once it meets all outstanding technical requirements.

Kind regards,

Salil Deo

Academic Editor

PLOS ONE

Additional Editor Comments (optional):

Reviewers' comments:

Reviewer's Responses to Questions

**Comments to the Author**

1. If the authors have adequately addressed your comments raised in a previous round of review and you feel that this manuscript is now acceptable for publication, you may indicate that here to bypass the “Comments to the Author” section, enter your conflict of interest statement in the “Confidential to Editor” section, and submit your "Accept" recommendation.

Reviewer #1: All comments have been addressed

Reviewer #2: All comments have been addressed

2. Is the manuscript technically sound, and do the data support the conclusions?

Reviewer #1: Yes

Reviewer #2: Yes

3. Has the statistical analysis been performed appropriately and rigorously? 

Reviewer #1: Yes

Reviewer #2: Yes

4. Have the authors made all data underlying the findings in their manuscript fully available?

Reviewer #1: Yes

Reviewer #2: Yes

5. Is the manuscript presented in an intelligible fashion and written in standard English?

Reviewer #1: Yes

Reviewer #2: Yes

6. Review Comments to the Author

Reviewer #1: Large single center study revealing outcomes of a procedure of great importance in our field. Revealing this paradigm may help with risk stratification and the intelligible risk profile obesity has the this epidemic increases.

Reviewer #2: The authors performed an extensive review of the article, according to the reviewers' suggestions. The English language significantly improved. The text was divided in appropriate sections that make the reading of the article clearer. However, I have to report the at the lines 181 and 234 are reported the patients with BMIs less than 18.5, but the authors excluded this group of patients in the revised analysis.

7. PLOS authors have the option to publish the peer review history of their article (what does this mean?). If published, this will include your full peer review and any attached files.

Reviewer #1: No

Reviewer #2: No

---

## [Editor Report · Acceptance letter]

8 Sep 2022

PONE-D-22-09560R1 

The Association Between Different Body Mass Index Levels and Midterm Surgical Revascularization Outcomes 

Dear Dr. Bagheri:

I'm pleased to inform you that your manuscript has been deemed suitable for publication in PLOS ONE. Congratulations! Your manuscript is now with our production department. 

Kind regards, 

on behalf of

Dr. Salil Deo 

Academic Editor

PLOS ONE